# Molecular mechanism establishing the OFF pathway in vision

**Florentina Soto** [1] ✉, **Chin-I Lin** [1,2,8], **Andrew Jo** [1,8], **Ssu-Yu Chou** [1], **Ellen G. Harding** [1], **Philip A. Ruzycki** [1], **Gail K. Seabold** [3], **Ronald S. Petralia** [3,4] & **Daniel Kerschensteiner** [1,5,6,7] ✉

Parallel ON and OFF (positive- and negative-contrast) pathways fundamental to vision arise at the complex synapse of cone photoreceptors. Cone pedicles form spatially segregated functionally opposite connections with ON and OFF bipolar cells. Here, we discover that mammalian cones express LRFN2, a cell-adhesion molecule, which localizes to the pedicle base. LRFN2 stabilizes basal contacts between cone pedicles and OFF bipolar cell dendrites to guide pathway-specific partner choices, encompassing multiple cell types. In addition, LRFN2 trans-synaptically organizes glutamate receptor clusters, determining the contrast preferences of the OFF pathway. ON and OFF pathways converge in the inner retina to regulate bipolar cell outputs. We analyze LRFN2's contributions to ON-OFF interactions, pathway asymmetries, and neural and behavioral responses to approaching predators. Our results reveal that LRFN2 controls the formation of the OFF pathway in vision, supports parallel processing in a single synapse, and shapes contrast coding and the detection of visual threats.

Visual systems from flies to humans process light increments and decrements in parallel ON and OFF pathways[1–4]. This dichotomy is fundamental to vision. It efficiently encodes naturalistic contrast distributions[2,5], enhances perceptual contrast sensitivity[3], and supports the construction of complex feature preferences (e.g., orientation and direction selectivity in the visual cortex)[6–9]. Dedicated OFF pathways detect approaching predators and elicit escapes[10,11].

In mammals, parallel ON and OFF pathways arise at the first synapse of the visual system, where cone pedicles form sign-conserving connections with OFF bipolar cells and sign-inverting connections with ON bipolar cells[12–15]. Cone pedicles release glutamate from synaptic ribbons, anchored at the top of narrow invaginations. ON bipolar cell dendrites enter these invaginations (~300 nm deep) and express metabotropic

glutamate receptors (mGluR6) on their tips[16–18]. Activation of mGluR6 closes cation-selective Trpm1 channels to invert the sign of the cone signal (OFF → ON)[16,19,20]. OFF bipolar cell dendrites contact the cone pedicle base between invaginations and localize ionotropic glutamate receptors (kainate and AMPA receptors) to postsynaptic specializations that keep the sign of the cone signal (OFF → OFF)[12,21–24]. Thus, parallel ON and OFF pathways arise at a single complex synapse.

Molecular mechanisms that shape pedicle invaginations, maintain contacts with ON bipolar cells, and align mGluR6 clusters with presynaptic release sites have been identified[16,25–31]. Yet, the mechanisms that establish basal contacts with OFF bipolar cells and cluster kainate and AMPA receptors in appropriate places on their dendrites remain unknown.

[1]Department of Ophthalmology and Visual Sciences, Washington University School of Medicine, St. Louis, MO, USA. [2]Graduate Program in Neuroscience, Division of Biological & Biomedical Sciences, Washington University School of Medicine, St. Louis, MO, USA. [3]Laboratory of Neurochemistry, National Institute on Deafness and Other Communication Disorders, National Institutes of Health, Bethesda, MD, USA. [4]Advanced Imaging Core, National Institute on Deafness and Other Communication Disorders, National Institutes of Health, Bethesda, MD, USA. [5]Department of Neuroscience, Washington University School of Medicine, St. Louis, MO, USA. [6]Department of Biomedical Engineering, Washington University School of Medicine, St. Louis, MO, USA. [7]Bright Center for Human Vision, Washington University School of Medicine, St. Louis, MO, USA. [8]These authors contributed equally: Chin-I Lin, Andrew Jo. ✉e-mail: sotof@wustl.edu; kerschensteinerd@wustl.edu

OFF bipolar cells comprise multiple types (six in mice) that differ in spatiotemporal and chromatic tuning and sensitivity to motion[23,32–36]. It is unclear whether the same cue controls the connectivity of cone pedicles with all OFF bipolar cell types (i.e., pathway-specific wiring) or if separate mechanisms establish connections with different partners in the OFF pathway (i.e., cell-type-specific wiring).

In addition to excitatory inputs to their dendrites, OFF bipolar cells receive inhibitory inputs to their axons[37,38]. This inhibition is mainly driven by ON-responsive amacrine cells, creating a push-pull system (OFF excitation-ON inhibition)[39–46]. How the system's two components interact to shape OFF bipolar cell signals is unclear.

Here, we discover that the leucine-rich repeat-containing cell-adhesion molecule LRFN2 is selectively expressed in cones across the mammalian clade, specifically at their basal synaptic contacts with OFF bipolar cells. LRFN2 maintains these contacts and clusters ionotropic glutamate receptors, ensuring robust dendritic transmission in the OFF pathway. Loss of *Lrfn2* abolishes direct excitatory input to OFF bipolar cells, leaving them reliant on inhibitory signals from the ON pathway for contrast encoding and weakening looming-driven defensive behaviors. These findings reveal a critical role for LRFN2 in guiding the formation and function of the OFF pathway in vision.

## Results

### *Lrfn2* expression in cones across species and development

The divergence of cone signals to ON and OFF bipolar cells is conserved across mammals[47,48]. To identify molecular mechanisms guiding the formation of the OFF pathway, we analyzed single-cell RNA-sequencing (scRNA-seq) data from the retinas of diverse species[48]. We identified cones and rods based on the expression of conserved marker genes (Supplementary Fig. 1). We searched for genes that were enriched in cones vs. rods because rods mostly form synapses with ON bipolar cells (in the form of a dedicated rod bipolar cell type)[49] and focused on LRR-containing cell-adhesion molecules, which play important roles in synapse development, including the formation of the retinal ON pathway[13,16,26,50–52]. Thus, we identified LRFN2 (also known as SALM1), a type I transmembrane protein, with six LRR domains, an immunoglobulin, and a fibronectin III domain in its extracellular N-terminus, and a PDZ domain in its intracellular C-terminus[53–55]. Across mammals (placental and marsupial), *Lrfn2* is expressed in cones and absent from rods; it shows limited expression in other retinal neurons (Fig. 1a). *Lrfn2* is missing from cones, or the selectivity of its expression is reduced in non-mammalian species (Fig. 1a).

Staining of retinal sections from adult mice with an antibody against LRFN2 labeled the outer plexiform layer (OPL) in a pattern

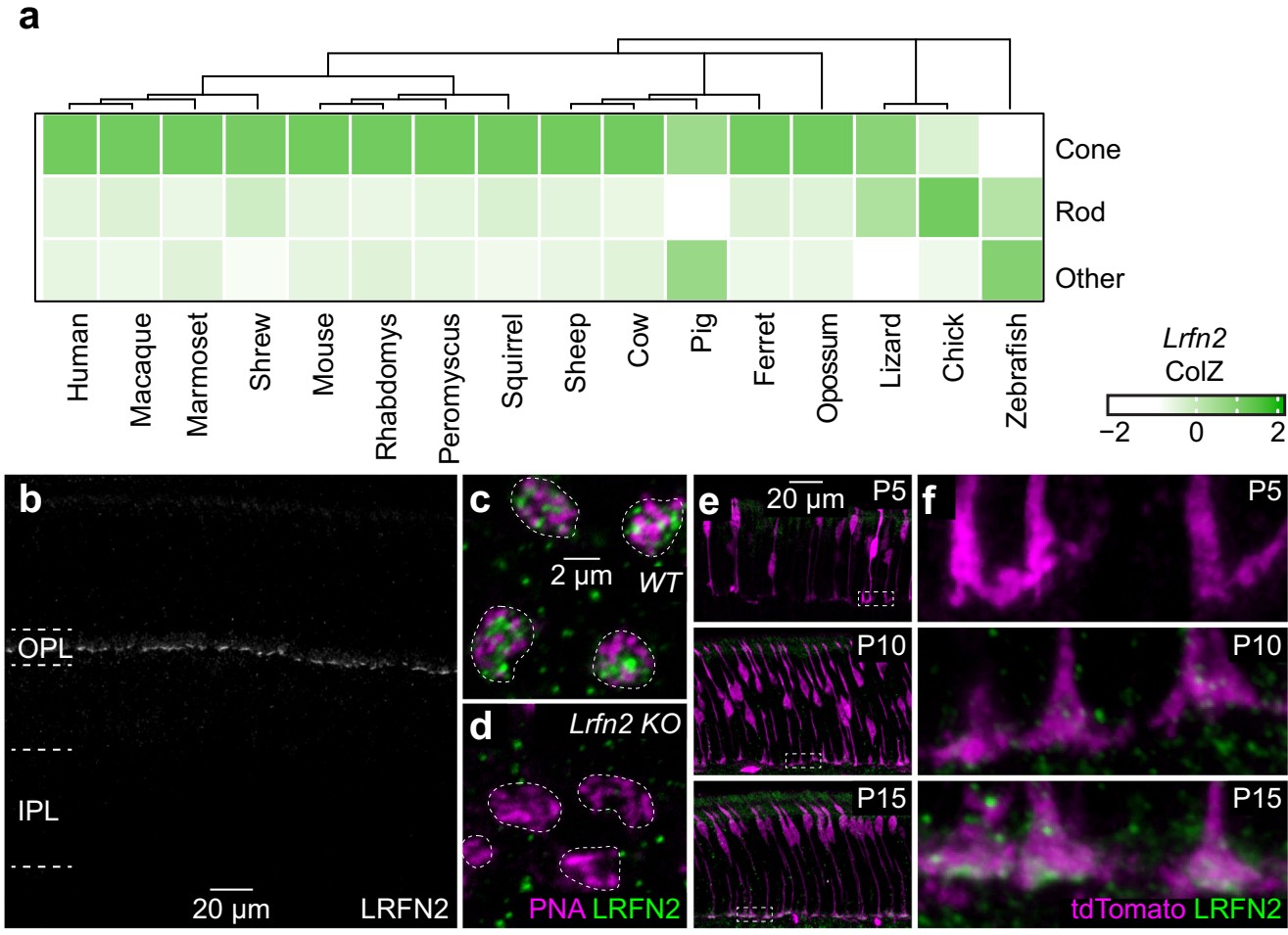

**Fig. 1 | *Lrfn2* expression across evolution and development. a** Heatmap of z-scores (by column) of scRNA-seq data[48] comparing expression of *Lrfn2* between cones, rods, and the other retinal neurons (averaging across all cell types) for a wide range of species. The phylogeny of these species is illustrated in the dendrogram above. **b** Representative image of a vibratome section of a P30 wild-type mouse stained for LRFN2. Similar images were obtained from five mice. **c, d** Representative images of the OPL of retinal flat mounts from wild-type (WT) and *Lrfn2 KO* mice stained for PNA (magenta) and LRFN2 (green). **e, f** Representative overview images (**e**, left) and zoomed-in excerpts (**f**, right) of retinal vibratome sections from P5 (top), P10 (middle), and P15 (bottom) *Opsin-tdTomato* mice stained for tdTomato (magenta) and LRFN2 (green). Similar images were obtained from at least three mice at each time point.

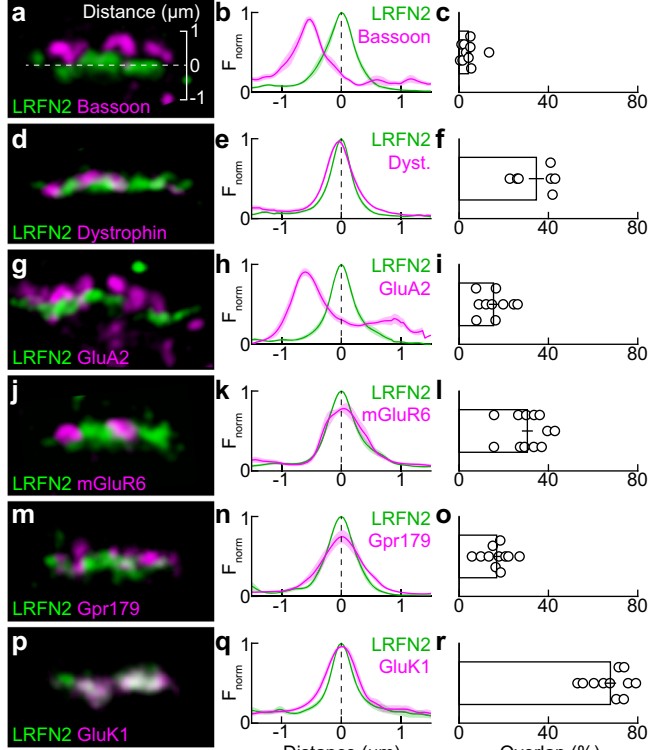

**Fig. 2 | LRFN2 localizes to the OFF compartment of cone synapses.**
**a** Representative super-resolution images of a cone pedicle synapse stained for
LRFN2 (green) and the ribbon-anchoring protein Bassoon (magenta). The scale bar
included in this image applies to all representative images in this figure.
**b** Normalized fluorescence intensity profiles of LRFN2 (green) and Bassoon
(magenta) staining orthogonal to the pedicle base ($n = 7$ cones from 3 retinas)
centered on the peak of the LRFN2 profile. Lines (shaded areas) indicate the mean
(±SEM). **c** Percentage overlap between LRFN2 and Bassoon signals ($n = 10$ cones
from 3 retinas). Bar (error bar) represents the mean (±SEM). **d**–**f** Analogous to (**a**–**c**)
for the stratification and overlap of LRFN2 (green) and Dystrophin (magenta, $n = 9$
[**e**] and 7 [**f**] cones from 3 retinas). **g**–**i** Analogous to (**a**–**c**) for the stratification and
overlap of LRFN2 (green) and GluA2 (magenta, $n = 9$ [**h**] and 10 [**i**] cones from 4
retinas). **j**–**l** Analogous to (**a**–**c**) for the stratification and overlap of LRFN2 (green)
and mGluR6 (magenta, $n = 11$ [**k**] and 12 [**l**] cones from 4 retinas). **m**–**o** Analogous to
(**a**–**c**) for the stratification and overlap of LRFN2 (green) and Gpr179 (magenta, $n = 9$
[**n**] and 11 [**o**] cones from 3 retinas). **p**–**r** Analogous to (**a**–**c**) for the stratification and
overlap of LRFN2 (green) and GluK1 (magenta, $n = 6$ [**q**] and 11 [**r**] cones from 4
retinas).

reminiscent of cone pedicles (Fig. 1b)[56]. Co-labeling of retinal flat
mounts for peanut agglutinin (PNA) lectin, a marker of cone active
zones, confirmed the localization of LRFN2 to cone pedicles (Fig. 1c).
This staining was absent in *Lrfn2* knockout (*KO*) mice, demonstrating
the specificity of the antibody (Fig. 1d).

The synapses between cones and bipolar cells develop between
postnatal day (P) 10 and P15, as mice open their eyes and the retina
transitions from generating spontaneous activity patterns (i.e., retinal
waves) to processing visual information[57]. We genetically labeled
cones by crossing a cell-type-specific Cre line[58] to a fluorescent
reporter strain[59], generating *Opsin-tdTomato* mice. We stained *Opsin-
tdTomato* retinas across development for LRFN2. LRFN2 became
detectable in cone pedicles between P10 and P15, consistent with a role
in synapse development (Fig. 1e, f).

## LRFN2 localizes to the OFF compartment of cone synapses
Cones establish functionally distinct connections with three partners
in a multi-compartment synapse. Cones release glutamate from rib-
bons anchored at the top of pedicle invaginations. Horizontal cell

dendrites are positioned immediately underneath the release sites,
express AMPA receptors (GluA2), and provide feedback to
cones[14,40,60–62]. ON bipolar cell dendrites enter invaginations and stra-
tify below horizontal cell dendrites, expressing metabotropic gluta-
mate receptors (mGluR6)[18,60]. Finally, OFF bipolar cell dendrites form
basal contacts with cone pedicles and cluster kainate (GluK1) and
AMPA (GluA1) receptors between invaginations[21,35,60,63].

We analyzed the position of LRFN2 in this complex synapse by
immunohistochemistry and super-resolution imaging. We quantified
the stratification of compartment-specific markers relative to LRFN2
and measured the overlap of the respective signals. Bassoon anchors
presynaptic ribbons and was located above LRFN2 (i.e., deeper in the
pedicle) with little overlap (Fig. 2a–c). Dystrophin, an actin-binding
protein of cone pedicles[64], was not vertically displaced from LRFN2 but
showed limited colocalization with its signal (Fig. 2d–f). GluA2, the
dominant glutamate receptor subunit of horizontal cells, localizes to
their dendrite tips and desmosome-like junctions below the cone
synapse[14,61,65]. GluA2 signals stratified above (on presumptive hor-
izontal cell dendrite tips) and below LRFN2 (at presumptive
desmosome-like junctions) with little overlap (Fig. 2g–i). Two com-
ponents of the ON bipolar cell postsynapse, mGluR6 and the orphan
receptor Gpr179[30,66,67], were not vertically separated but overlapped
little with LRFN2 (Fig. 2j–o). By contrast, GluK1, the postsynaptic
receptor of most OFF bipolar cells, co-stratified and co-localized
extensively with LRFN2 (Fig. 2p–r).

Our colocalization experiments detected LRFN2 with an antibody
against its extracellular N-terminus. To confirm that LRFN2 is expres-
sed in cones (as the scRNA-seq data[48] in Fig. 1 suggests), we measured
the vertical position of LRFN2 relative to GluK1 with an antibody
against the intracellular C-terminus of LRFN2. Whereas the N-terminal
LRFN2 signal co-stratified with GluK1, the C-terminal LRFN2 signal
stratified above GluK1 (i.e., within the cone pedicle, Supplementary
Fig. 2). Together, these data indicate that LRFN2 is expressed in cones,
where it selectively localizes to basal contacts with OFF bipolar cells
opposite ionotropic glutamate receptors clusters.

## LRFN2 controls pedicle size and compartment-specific synapse assembly
To explore the contributions of LRFN2 to synapse development and
visual processing, we obtained *Lrfn2 KO* mice. Staining for cell-type-
specific markers revealed preserved cell body positions and neurite
stratification patterns in *Lrfn2 KO* retinas (Supplementary Fig. 3).
Cones were present in normal numbers (Fig. 3a–c), but the size of the
cone pedicles and active zones was reduced in *Lrfn2 KO* mice com-
pared to wild-type littermates (Fig. 3a, b, d, e), suggesting a loss of
connections.

We stained for markers to test whether this loss was compartment-
specific or equally affected connections with all postsynaptic partners
(i.e., horizontal cells, ON bipolar cells, and OFF bipolar cells). Bassoon
and Dystrophin retained their regular presynaptic positions in *Lrfn2 KO*
compared to wild-type mice (Fig. 4a–f); the bistratified pattern of GluA2
on horizontal cell dendrite tips and desmosome-like junctions was pre-
served (Fig. 4g, h); and the expression of mGluR6 and Gpr179 in the ON
bipolar cell postsynapse was unchanged (Fig. 4i, j). However, GluK1 and
GluA1, enriched in the wild-type OFF bipolar cell postsynapse, failed to
cluster in *Lrfn2 KO* mice (Fig. 4m–p).

The images in Fig. 4 were acquired with the same settings after
identical staining in wild-type and *Lrfn2 KO* retinas to allow for com-
parisons of receptor expression and clustering. GluK1 and GluA1 sig-
nals became visible in *Lrfn2 KO* retinas when we increased the laser
power and detector voltage. However, these weaker, diffuse signals
were displaced vertically below the pedicle base, losing their close
apposition with presynaptic markers (Supplementary Fig. 4). *Grik1*
(encoding GluK1) and *Gria1* (encoding GluA1) mRNA expression were
unaffected by LRFN2 removal (Supplementary Fig. 5) as were bipolar

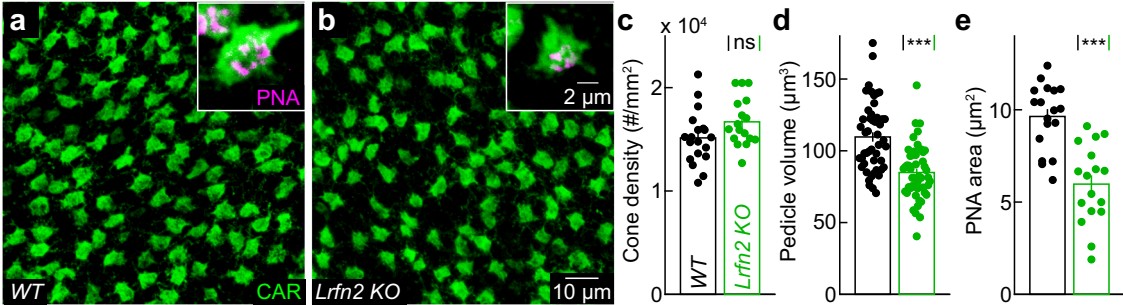

**Fig. 3 | LRFN2 regulates cone pedicle size. a, b** Representative images of retinal flat mounts from wild-type (**a**) and *Lrfn2 KO* (**b**) mice stained for cone arrestin (CAR, green). Inserts show zoomed-in views of individual cone pedicles co-stained for the active zone marker PNA (magenta). **c** Summary data on the density of cones in wild-type (black, $n = 19$ retinas) and *Lrfn2 KO* mice (green, $n = 17$ retinas, $p = 0.08$ by two-sided Mann–Whitney *U* test). In (**c**–**e**), the bars (error bars) represent the mean

(±SEM). **d** Summary data on the pedicle volume in wild-type (black, $n = 48$ cones from 22 retinas) and *Lrfn2 KO* retinas (green, $n = 50$ cones from 24 retinas, $p = 6.9 \times 10^{-7}$ by two-sided Mann–Whitney *U* test). **e** Summary data of the PNA area in maximum intensity projections of cones pedicles from wild-type (black, $n = 18$ cones from 10 retinas) and *Lrfn2 KO* retinas (green, $n = 17$ cones from 9 retinas, $p = 3.4 \times 10^{-5}$ by two-sided Mann–Whitney *U* test).

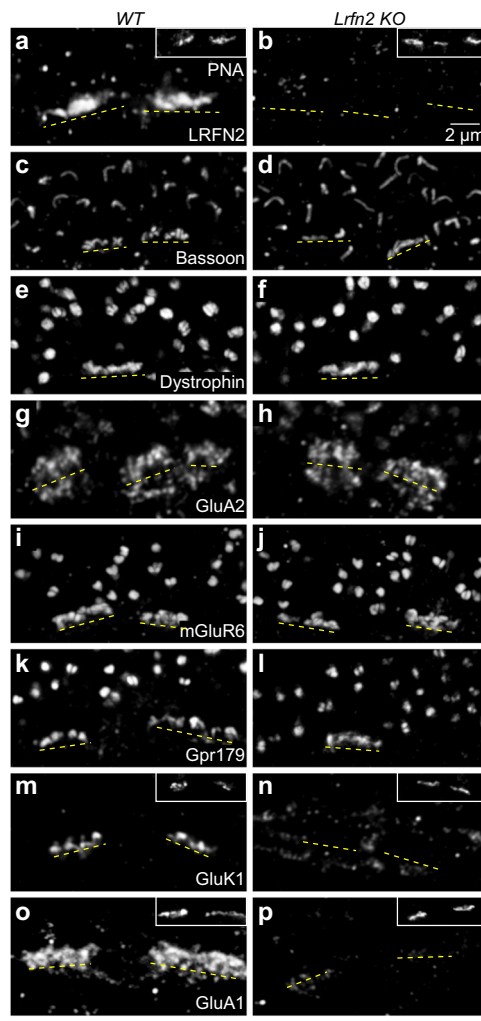

**Fig. 4 | LRFN2 controls compartment-specific synapse assembly.**
**a**, **b** Representative super-resolution images of the OPL stained for LRFN2 in wild-type (**a**) and *Lrfn2 KO* (**b**) mice. Dashed yellow lines indicate the positions of cone pedicles. Insets confirm cone positions by PNA staining. **c**–**p** Analogous to (**a**, **b**) for Bassoon (**c**, **d**), Dystrophin (**e**, **f**), GluA2 (**g**, **h**), mGluR6 (**i**, **j**), Gpr179 (**k**, **l**), GluK1 (**m**, **n**), and GluA1 (**o**, **p**) staining. For each staining, similar images to those shown here were acquired from at least three mice.

cell numbers (Supplementary Fig. 6). Thus, LRFN2 selectively controls the molecular assembly and stratification of the OFF compartment of the cone synapse.

### LRFN2 maintains basal contacts between cone pedicles and OFF bipolar cell dendrites
The vertical displacement of the diffuse GluK1 and GluA1 staining in *Lrfn2 KO*, compared to the dense receptor clusters in wild-type retinas, could reflect differences in the receptor localization within bipolar cell dendrites and/or a loss of contacts between OFF bipolar cell dendrites and cone pedicles. We crossed *Vsx1-cerulean* mice, in which OFF bipolar cells (types 1–4) are sparsely labeled[37,68], to *Lrfn2 KO* mice to analyze contacts between bipolar cell dendrites and cone pedicles.

OFF bipolar cell dendrites correctly targeted the OPL (and OFF bipolar cell axons the inner plexiform layer or IPL) in *Lrfn2 KO* and wild-type mice (Fig. 5a, b). We co-labeled retinal flat mounts for PNA. Whereas the signal of OFF bipolar cell dendrites overlapped with cone active zones in wild-type retinas (Fig. 5c, e; peak offset: $-0.06 \pm 0.25 \,\mu m$, $n = 9$ bipolar cells from 3 retinas), OFF bipolar cell dendrites were detached below the cones in *Lrfn2 KO* mice (Fig. 5d, f; peak offset: $-0.71 \pm 0.10 \,\mu m$, $n = 8$ bipolar cells from 3 retinas, $p = 4.6 \times 10^{-5}$ for wild-type vs *Lrfn2 KO* by Mann–Whitney U test). Thus, LRFN2, in addition to controlling postsynaptic receptor clustering, maintains basal contacts between OFF bipolar cell dendrites and cone pedicles.

### LRFN2 is required for signal transmission from cones to OFF bipolar cells
Our anatomical data suggested that LRFN2 guides the structural assembly of connections between cones and OFF bipolar cells. To determine LRFN2's contributions to synaptic function, we targeted bipolar cells in the inner nuclear layer of retinal flat mounts for whole-cell patch-clamp recordings (Fig. 6a, f). We included a fluorescent dye (Alexa 488) in the pipette solution and acquired two-photon image stacks at the end of each recording to reconstruct bipolar cells and classify them as ON or OFF based on axonal stratification patterns[69].

Bipolar cells have excitatory synaptic conductances in their dendrites (ionotropic glutamate receptors in OFF and Trpm1 channels in ON bipolar cells) and inhibitory receptors in their axons (GABA and glycine receptors in ON and OFF bipolar cells)[15,47]. We isolated excitatory and inhibitory currents via the holding potential ($-60 \,mV$ and $0 \,mV$, respectively) in voltage-clamp recordings. We presented square-wave-modulated spots of light (60 µm diameter, 2 s ON and 2 s OFF) centered on the recorded cell. In wild-type retinas, OFF bipolar cells received strong excitation at light OFF and inhibition at light ON (i.e., push-pull system, Fig. 6a–e). Excitatory inputs to OFF bipolar cells were

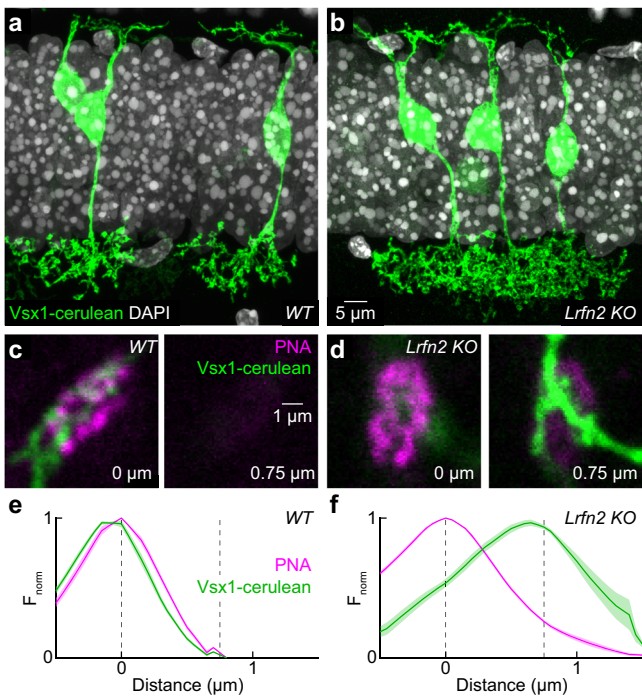

**Fig. 5 | LRFN2 maintains basal cone contacts with OFF bipolar cell dendrites.**
**a, b** Representative super-resolution images of retinal vibratome sections from *Vsx1-cerulean* (green) sections on wild-type (**a**) and *Lrfn2 KO* (**b**) backgrounds stained for DAPI (gray). **c, d** Representative image pairs of *Vsx1-cerulean* (green) retinal flat mounts on wild-type (**c**) and *Lrfn2 KO* (**d**) backgrounds stained for PNA (magenta). Images are vertically separated by 0.75 μm. **e, f** Normalized fluorescence intensity profiles from cone pedicles in *Vsx1-cerulean* (green) retinal flat mounts on wild-type (**e**, *n* = 10 cones from 3 retinas) and *Lrfn2 KO* (**f**, *n* = 8 cones from 3 retinas) backgrounds stained for PNA (magenta). Lines (shaded areas) indicate the mean (±SEM).

drastically reduced in *Lrfn2 KO* retinas, while inhibition was unchanged (i.e., pull-only system, Fig. 6a–e).

ON bipolar cells exhibited a similar push-pull arrangement of synaptic inputs, featuring ON excitation and OFF inhibition (Fig. 6f–j). Additionally, ON bipolar cells receive substantial tonic excitation, which is diminished at light OFF (Fig. 6g and Supplementary Fig. 7). Neither excitation, inhibition, nor their modulation by light showed significant differences between ON bipolar cells in *Lrfn2 KO* and wild-type retinas (Fig. 6f–j and Supplementary Fig. 7).

Electroretinography (ERG) allows in vivo measurements of cone-driven ON bipolar cell responses (i.e., b-waves). Consistent with our patch-clamp results, we observed no differences between the ERG b-wave responses of *Lrfn2 KO* and wild-type mice to flickering stimuli across a range of frequencies (Supplementary Fig. 8). We performed these measurements in light-adapted retinas and on a *Gnat1 KO* background to eliminate contributions from rods (Supplementary Fig. 8)[70]. Together, these results demonstrate that LRFN2 is required to transmit cone signals to OFF bipolar cells without influence on the ON pathway arising at the same synapse.

### Uniform action of LRFN2 across the OFF pathway
The OFF pathway encompasses six bipolar cell types, which differ in their spatiotemporal and chromatic stimulus preferences and motion sensitivity and stratify their axons at different depths of the IPL[23,32–36]. We used two-photon glutamate imaging to analyze the bipolar cell output across the IPL[21,33]. We expressed iGluSnFR via an adeno-associated virus targeting most retinal cell types to generate a glutamate-sensing neurite matrix in the IPL (Fig. 7a). In this matrix, we identified regions of interest (ROIs) capturing release from individual boutons of bipolar cell axons by spatiotemporal signal correlations with a previously described algorithm (Fig. 7a)[33]. We presented an achromatic chirp stimulus in which the intensity of a light spot (100 μm diameter) was stepped up and down (1.5 s ON, 1.5 s OFF), followed by sinusoidal modulations of increasing temporal frequency (0.5–40 Hz, 100% contrast) and amplitude (10–100% contrast, 1 Hz).

Plotting the average polarity index (±SEM, see "Methods") of ROI responses to the light steps revealed an abrupt transition from OFF (polarity ˜−1) to ON preferences (polarity ˜1) halfway through the depth of the IPL (Fig. 7b)[21,33], matching the stratification patterns of OFF and ON bipolar cell axons[69]. Across the chirp stimulus, glutamate release from OFF bipolar cells was remarkably similar between wild-type and *Lrfn2 KO* mice (Fig. 7c) despite the loss of dendritic excitation in the latter (Fig. 6). This suggests that axonal inhibition is sufficient to modulate glutamate release from OFF bipolar cells (pull-only system)[21]. OFF bipolar cells receive inhibition from interneurons driven by ON bipolar cells[40–43,46]. We applied L-APB, an agonist of mGluR6 receptors, to silence the ON pathway. In the presence of L-APB, glutamate release from OFF bipolar cells persisted in wild-type retinas (Fig. 7d, g, h), demonstrating that dendritic excitation is sufficient to drive release (push-only system). Contrast encoding in the OFF bipolar cell output became more linear in L-APB, evidenced by the suppression of the iGluSnFR signal below baseline during the ON phases of the chirp stimulus (Fig. 7c, d).

In stark contrast to wild-type retinas, light-evoked glutamate release from OFF bipolar cells in *Lrfn2 KO* retinas was blocked by L-APB; signal reliability across stimulus repeats and entrainment to stimulus fluctuations was lost (Fig. 7d, i, j). The L-APB effects were uniform across the depths of the IPL occupied by OFF bipolar cell axons (Fig. 7b, i, j, Supplementary Figs. 9–11). Glutamate release from ON bipolar cells was suppressed in wild-type and *Lrfn2 KO* retinas, confirming the efficacy of L-APB (Fig. 7e–j). In the presence of L-APB, faint OFF responses emerged in the ON sublamina, particularly in wild-type mice (Fig. 7f). These responses were strongest near the center of the IPL and gradually diminished toward the periphery, suggesting they reflect spillover of glutamate release from OFF bipolar cells (Supplementary Fig. 9). Thus, dendritic excitation (push-only) and axonal inhibition (pull-only) are sufficient to drive light-evoked glutamate release from OFF bipolar cells; axonal inhibition rectifies the output of OFF bipolar cells[21,41]; and dendritic excitation across all OFF bipolar cell types depends on LRFN2.

### LRFN2 contributes to visual threat detection and innate defensive behaviors
Mammals, from mice to humans, exhibit innate defensive responses to expanding shadows (i.e., looming stimuli) that signal objects on a collision course and approaching predators[71,72]. The retinal circuits that detect looming stimuli and elicit defensive responses center on the OFF pathway and, in mice, relay signals to the brain via transient OFFα ganglion cells (tOFFα cells)[10,11,73]. We targeted tOFFα cells for patch-clamp recordings under infrared illumination based on their large soma size (Fig. 8a)[74]. We filled cells with a fluorescent dye (Alexa488) in the pipette solution and confirmed our targeting by two-photon imaging analyses of their dendritic morphology at the end of each recording[74]. The looming responses of tOFFα ganglion cells were reduced in *Lrfn2 KO* compared to wild-type mice (Fig. 8b, c) in control solutions. Furthermore, L-APB nearly abolished looming responses in *Lrfn2 KO* retinas but had little effect in wild-type retinas (Fig. 8d, e).

When mice were placed in a behavioral arena with virtual shelters on two sides (i.e., areas not illuminated by the monitor in the arena ceiling), they fled to a shelter and froze in response to looming stimuli (Fig. 8f–h)[10]. Although there was a trend toward less freezing in *Lrfn2 KO* mice, their responses were not significantly different from wild-type littermates (Fig. 8g, h). We wanted to block the ON pathway to test the behavioral contribution of dendritic OFF bipolar cell excitation and

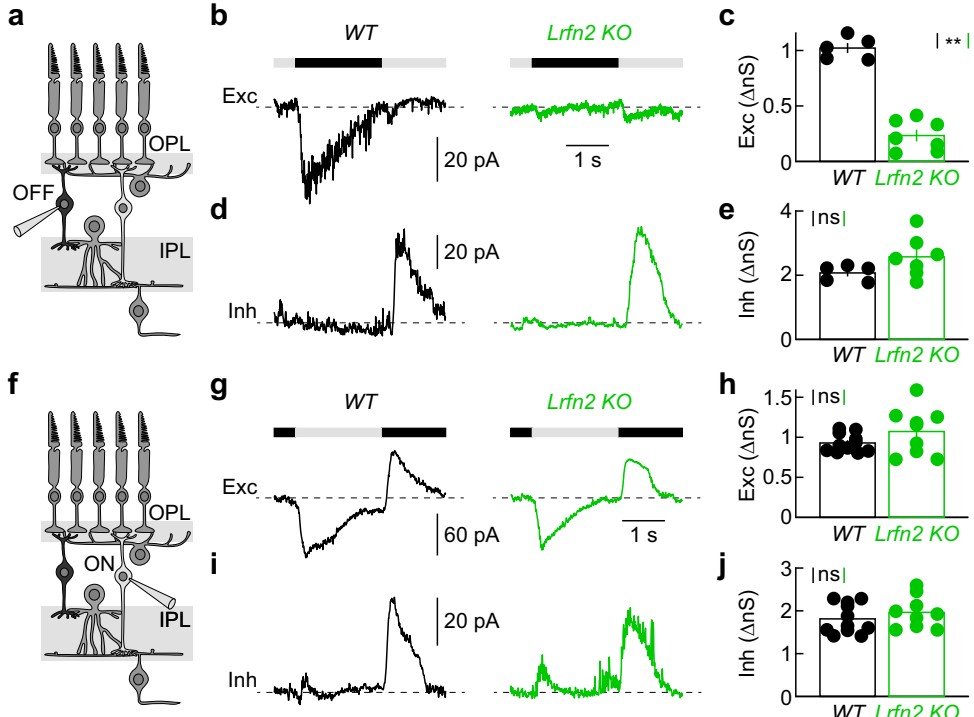

**Fig. 6 | LRFN2 supports signal transmission from cones to OFF bipolar cells.**
**a** Schematic illustrating the retinal circuitry and OFF bipolar patch-clamp record-
ings. The schematic shows a cross-sectional retina view; OFF bipolar cells were
targeted in retinal flat mounts. **b** Representative EPSC responses to a square-wave-
modulated (2 s ON and 2 s OFF) 60 μm spot of light recorded from OFF bipolar cells
in wild-type (black, left) and *Lrfn2 KO* (green, right) mice. Stimulus timing is illu-
strated above response traces. **c** Summary data of excitatory synaptic con-
ductances during the stimulus OFF phase (wild-type: *n* = 5 cells from 5 retinas, *Lrfn2
KO*: *n* = 7 cells from 7 retinas, *p* = 0.0025 by two-sided Mann–Whitney *U* test). Bar
(error bar) indicates the mean (±SEM). **d, e** Representative IPSC responses (**d**) and
summary data (**e**) of inhibitory synaptic conductances during the stimulus ON
phase (wild-type: *n* = 5 cells from 5 retinas, *Lrfn2 KO*, green: *n* = 7 cells from 7
retinas, *p* = 0.34 by two-sided Mann–Whitney *U* test). Bar (error bar) indicates the

mean (±SEM). **f** Schematic illustrating the retinal circuitry and ON bipolar patch-
clamp recordings. **g** Representative EPSC responses to a square-wave-modulated
(2 s ON and 2 s OFF) 60 μm spot of light recorded from ON bipolar cells in wild-type
(left, black) and *Lrfn2 KO* (right, green) mice. Stimulus timing is illustrated above
response traces. **h** Summary data of excitatory synaptic conductances during the
stimulus ON phase (wild-type, black: *n* = 11 cells from 11 retinas, *Lrfn2 KO*, green:
*n* = 9 cells from 9 retinas, *p* = 0.32 by two-sided Mann–Whitney *U* test). Bars (error
bars) represent the mean (±SEM). **i, j** Representative IPSC responses (**i**) and sum-
mary data (**j**) of inhibitory synaptic conductances during the stimulus OFF phase
(wild-type, black: *n* = 11 cells from 11 retinas, *Lrfn2 KO*, green: *n* = 9 cells from 9
retinas, *p* = 0.36 by two-sided Mann–Whitney *U* test). Bars (error bars) indicate the
mean (±SEM).

the role of LRFN2. We injected different concentrations of L-APB
(100 μM and 1 mM, 1 μL) intraocularly 30 min before recording ERGs.
Whereas 1 mM L-APB blocked both the a- and b-waves, reflecting
photoreceptor hyperpolarization and ON bipolar cell depolarization,
respectively, 100 μM L-APB selectively blocked the b-wave (Supple-
mentary Fig. 12). We therefore injected 100 μM L-APB binocularly
30 min before testing the looming responses of wild-type and *Lrfn2 KO*
mice. While the looming responses of wild-type mice were unaffected
by L-APB injections, freezing was drastically reduced in *Lrfn2 KO* mice.
We cannot rule out that LRFN2 influences looming responses through
visual circuits downstream of the retina. However, the limited
expression of *Lrfn2* in the respective brain areas (including the
superior colliculus, parabigeminal nucleus, and dorsal periaqueductal
gray) and the effect of intraocular L-APB injections argue against
this[53,75–77].

Together, these results demonstrate the importance of LRFN2 for
dendritic excitation of OFF bipolar cells and elucidate its contributions
to looming responses in the retinal output and the defensive behaviors
elicited by these responses.

## Discussion

Sensory systems parse information into parallel pathways that extract
different features of the environment. Insight into the molecular
mechanisms that establish parallel pathways is fundamental to under-
standing sensory system development and critical for therapeutic

approaches to restore sensory pathways disrupted by disease and injury
(e.g., vision restoration by photoreceptor replacement)[78]. In the visual
system, the divergence of cone signals into ON and OFF bipolar cell
pathways underlies efficient retinal encoding of naturalistic contrast
distributions[2,5], supports the construction of complex feature
preferences[6–9], and facilitates the rapid detection of visual threats[10,11].
This divergence occurs at the cone pedicle–a uniquely complex
synapse at which cone outputs are segregated into invaginating con-
tacts with ON bipolar cells and basal contacts with OFF bipolar cells[12–15].
Glutamate released from presynaptic ribbons at the top of these inva-
ginations diffuses varying distances to metabotropic (ON) and iono-
tropic (OFF) receptors[17,18,21,23,24,79].

Here, we report that LRFN2 is expressed in cones (but not rods)
across a broad range of mammals (Fig. 1). Within cones, LRFN2 localizes
selectively to the pedicle base (Fig. 2). Genetic deletion of LRFN2 does
not affect invaginating contacts with ON bipolar cells[56] but eliminates
basal connections with OFF bipolar cells (Figs. 4–6). In the OFF com-
partment, LRFN2 maintains physical contact with bipolar cell dendrites
(Fig. 5) and organizes kainate and AMPA receptor clusters opposite the
pedicle base (Fig. 4). These functions of LRFN2 closely resemble those
of ELFN1 and ELFN2 in the ON pathway, which ensure the presence of
ON bipolar cell dendrites in cone pedicle invaginations and cluster
mGluR6 receptors in their postsynaptic membranes[25,80]. Like LRFN2,
ELFN1 and ELFN2 are type I transmembrane proteins with extracellular
LRR domains[81]. Thus, two sets of presynaptic LRR-containing cell

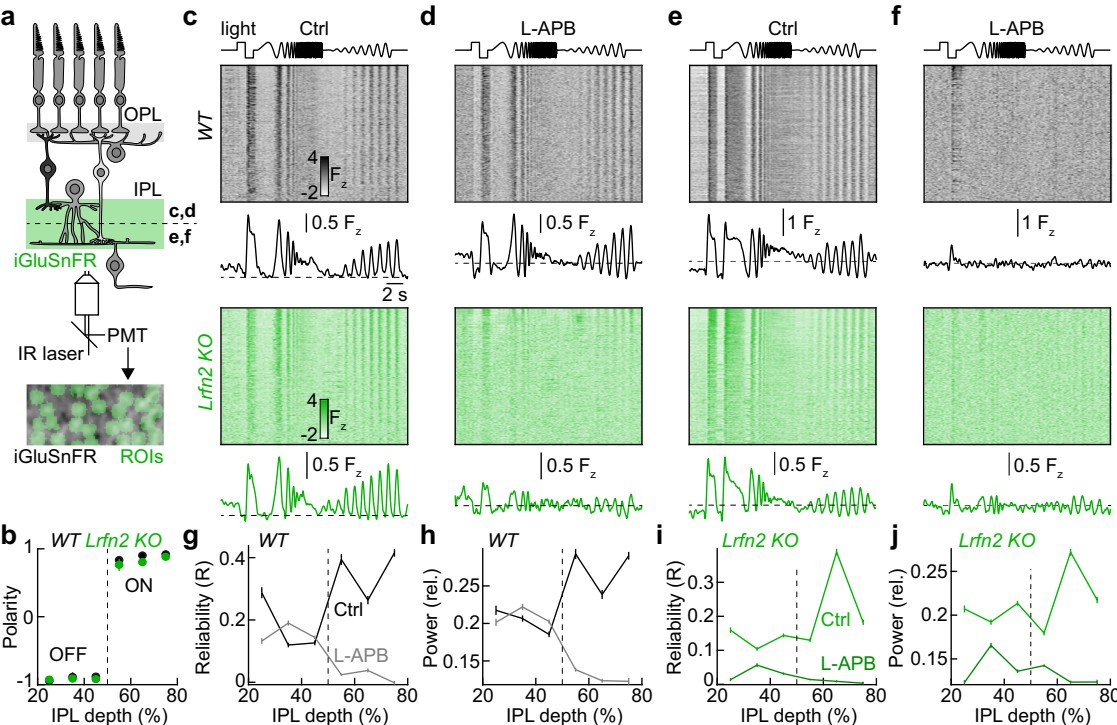

**Fig. 7 | LRFN2 shapes bipolar cell output across the OFF pathway. a** Schematic illustrating our two-photon glutamate imaging approach in the IPL. We virally expressed iGluSnFR (green) in most retinal neurons, generating a glutamate-sensitive neurite matrix in the IPL[3,4]. In this matrix, we segmented regions of interest (ROIs, green), capturing glutamate release from individual bipolar cell boutons based on spatiotemporal signal correlations[4]. **b** Summary data (mean ± SEM) of polarity indices at different IPL depths in wild-type (black, *n* = 3 retinas) and *Lrfn2 KO* (green, *n* = 3 retinas) mice. **c** Top trace illustrates the intensity trajectory of the chirp stimulus presented in a spot (diameter: 100 μm). Heatmaps and response traces (mean ± SEM) show ROI responses to the stimulus in the OFF sublamina (i.e., IPL depth < 50%) of wild-type (upper panels, gray, *n* = 464 ROIs from 3 retinas) and *Lrfn2 KO* (lower panels, green, *n* = 1001 ROIs from 3 retinas) retinas in mACSF$_{NaHCO3}$

(i.e., Ctrl). **d** Analogous to (**c**) for recordings from wild-type (upper panels, gray, *n* = 496 ROIs from 3 retinas) and *Lrfn2 KO* (lower panels, green, *n* = 486 ROIs from 3 retinas) retinas in mACSF$_{NaHCO3}$ with L-APB (20 μM). **e, f** Analogous to (**c, d**) for ROIs in the IPL ON sublamina (i.e., IPL depth > 50%) in mACSF$_{NaHCO3}$ (**e**, wild-type: *n* = 340 ROIs from 3 retinas, *Lrfn2 KO*: *n* = 866 ROIs from 3 retinas) and mACSF$_{NaHCO3}$ with L-APB (**f**, wild-type: *n* = 249 ROIs from 3 retinas, *Lrfn2 KO*: *n* = 714 ROIs from 3 retinas). **g, h** Summary data (mean ± SEM) of the repeat reliability (**g**) and power at the stimulus frequency (i.e., 1 Hz) during the increasing-contrast segment of the chirp stimulus (**h**) of wild-type ROI responses in mACSF$_{NaHCO3}$ (Ctrl, black, *n* = 3 retinas) and mACSF$_{NaHCO3}$ with L-APB (gray, *n* = 3 retinas). **i, j** Analogous to (**g, h**) for recordings from *Lrfn2 KO* retinas (Ctrl, light green, *n* = 3 retinas; L-APB, dark green, *n* = 3 retinas).

surface proteins partition into separate compartments of cone pedicles to maintain physical contact with dendrites of different bipolar cell types and organize clusters of specific postsynaptic receptors (mGluR6 in ON and kainate/AMPA receptors in OFF bipolar cells) that determine the functional divergence of ON and OFF pathways. In both cases, the presynaptic cues guide pathway-specific partner choices, establishing connections with multiple cell types.

Whether LRFN2 mediates synaptic adhesion with OFF bipolar cells and postsynaptic receptor clustering independently or if one deficit in *Lrfn2 KO* mice results from the other (e.g., glutamate receptors diffuse because synaptic adhesions are lost) remains to be explored. Similarly, the molecular mechanisms by which LRFN2 exerts trans-synaptic control over OFF bipolar cells are unknown. No trans-synaptic interaction partners of LRFN2 have yet been identified[82,83]. In cultured cells, the extracellular region of LRFN2 can interact with NMDA and AMPA receptors in cis (i.e., in the same membrane)[55,84–87], and in the ON pathway, ELFN1 and ELFN2 interact trans-synaptically with mGluR6[25,80,81]. While it is possible that LRFN2 at the cone pedicle base interacts trans-synaptically with AMPA and kainate receptors in OFF bipolar cells, in preliminary co-immunoprecipitation experiments, we did not detect such interactions. Alternatively, LRFN2 could control synaptic adhesion and postsynaptic receptor clustering through yet-unidentified trans-synaptic partners[35,48,88] or, indirectly, through presynaptic actions[89].

In the ON pathway, additional surface proteins (some containing LRR domains) that regulate the molecular architecture of the pre- and postsynapse have been identified[13,16]. It remains to be uncovered

whether similar diversity exists at the origins of the OFF pathway and, if so, to what end.

Our findings align with Hasan and Gregg[56], who also located LRFN2 at the cone pedicle base. However, while Hasan and Gregg[56] focused on the ON pathway—finding subtle changes in the ERG b-wave —we detected no differences in the ERG (Supplementary Fig. 8) or synaptic inputs from cones to ON bipolar cells in *Lrfn2 KO* mice (Fig. 5 and Supplementary Fig. 7). Instead, our anatomical, physiological, and behavioral data (Figs. 2 and 4–8) reveal that LRFN2 is specifically required for synapses between cones and OFF bipolar cells, a pathway that Hasan and Gregg[56] did not investigate.

In the hippocampus (and other brain areas), LRFN2 is found in the postsynaptic densities of excitatory synapses[53,55,84]. There is debate about the deficits in hippocampal synapse development of *Lrfn2 KO* mice[85,90], but overall, phenotypes of LRFN2 disruption appear milder in the hippocampus than we observed in the retina, affecting similar domains (e.g., receptor clustering, synaptic adhesion, and synapse morphology)[53,55,85,90]. Milder phenotypes may indicate compensation by other mechanisms. The opposite localization of LRFN2 (retina: presynaptic, hippocampus: postsynaptic) matches observations for other cell-adhesion molecules and components of the intracellular synaptic scaffold (including PSD95)[13,14,16,80]. The purpose of this partially inverse molecular architecture of the photoreceptor synapse remains to be fully understood.

In *Lrfn2 KO* mice, OFF bipolar cells lack dendritic excitation. Still, they encode luminance contrast in their axonal output (Fig. 7,

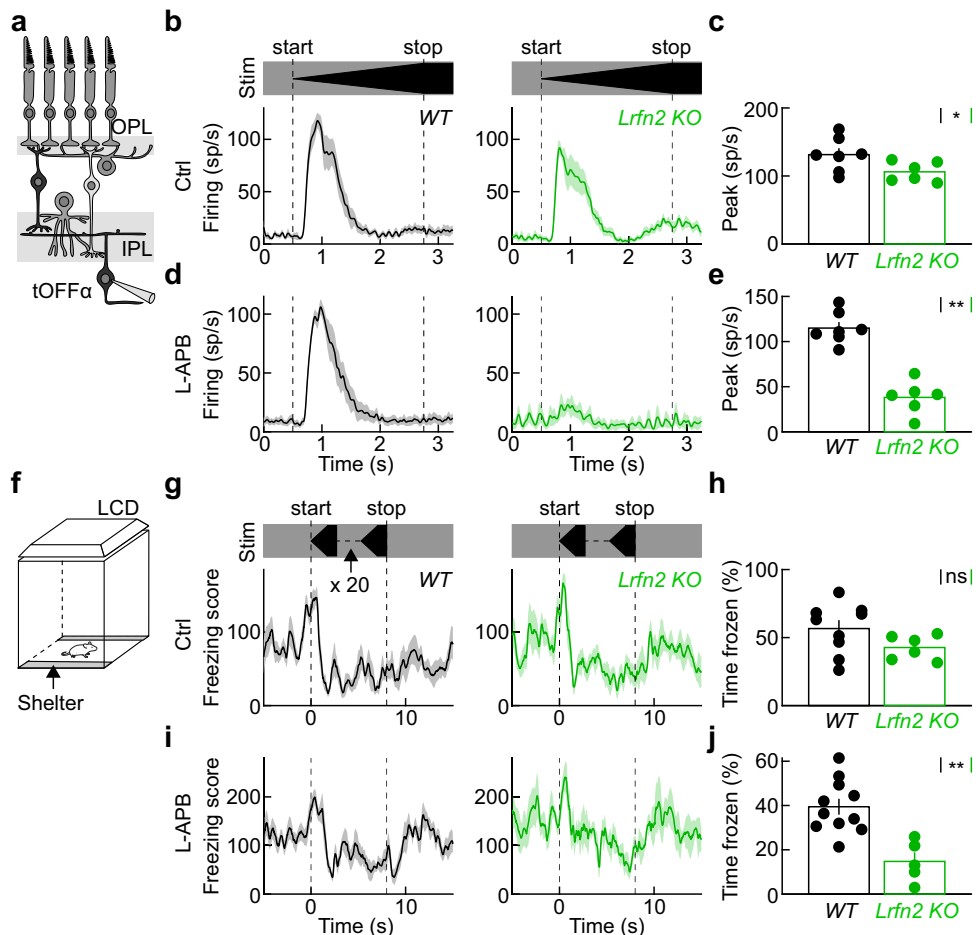

**Fig. 8 | LRFN2 contributes to visual threat detection and innate defensive behaviors. a** Schematic illustrating the retinal circuitry and tOFFα ganglion cell patch-clamp recordings. The schematic shows a cross-sectional retina view; tOFFα ganglion cells were targeted in retinal flat mounts. **b, d** Traces (shaded regions) show average (±SEM) spike responses of tOFFα ganglion cells in wild-type (left panel, black) and *Lrfn2 KO* (right panel, green) retinas in mACSF$_{NaHCO3}$ with (**d**) and without (**b**) L-APB (20 mM). **c, e** Summary data of tOFFα ganglion cell peak response amplitudes recorded in mACSF$_{NaHCO3}$ (**c**, wild-type, black: *n* = 7 cell from 7 retinas, *Lrfn2 KO*, green: *n* = 6 cells from 6 retinas, *p* = 0.035 by two-sided Mann–Whitney *U* test) and mACSF$_{NaHCO3}$ with L-APB (**e**, n's identical to ctrl, *p* = 0.0012 by two-sided

Mann–Whitney *U* test). Bars (error bars) represent the mean (±SEM). **f** Schematic of the behavioral arena with virtual shelters on two sides and an LCD monitor in the ceiling, on which looming stimuli were presented when mice crossed the center of the arena. **g, h** Freezing score response traces (**g**, mean ± SEM) and summary data (**h**, i.e., the fraction of the time frozen from stimulus onset to 10 s later) of wild-type (black, *n* = 9 mice) and *Lrfn2 KO* (green, *n* = 6 mice, *p* = 0.18 by two-sided Mann–Whitney *U* test) mice to looming stimuli. Bars (error bars) indicate the mean (±SEM). **i, j** Analogous to (**g, h**) for wild-type (black, *n* = 11 mice) and *Lrfn2 KO* (green, *n* = 5 mice, *p* < 0.0018 by two-sided Mann–Whitney *U* test) mice intraocularly injected with L-APB (100 μM).

Supplementary Figs. 9–11). This highlights the sufficiency of axonal inhibition (i.e., the pull part of the push-pull system) in modulating glutamate release. The modus operandi of OFF bipolar cells in *Lrfn2 KO* mice resembles their function in dim light, near the threshold of vision, when rod signals reach OFF bipolar cells solely through axonal inhibition[91,92]. Furthermore, one OFF bipolar cell type, the glutamatergic monopolar interneuron (GluMI or type 1B cell), loses its dendrites during development and is entrained to light by axonal inhibition at all light levels[35,93].

Although ON bipolar cells also receive OFF inhibition, this input alone cannot drive glutamate release (Fig. 7, Supplementary Figs. 9–11)[21]. In addition, whereas ON inhibition rectifies the output of OFF bipolar cells, glutamate release from ON bipolar cells encodes contrast linearly, even in the presence of OFF inhibition (Fig. 7, Supplementary Figs. 9–11)[21]. This asymmetry between the ON (linear) and OFF (rectified) pathways is reflected in the retinal output (i.e., the spike trains of retinal ganglion cells), and optimizes efficient coding of naturalistic contrast distribution[2,94–96]. The dedicated retinal OFF pathway has also been hypothesized to aid the detection of approaching predators. Consistent with this hypothesis, we find that LRFN2 shapes the

responses of tOFFα ganglion cells and the innate defensive behaviors relying on these responses (Fig. 8)[10,11].

## Methods

### Animals

*Lrfn2 KO* mice were obtained from the Mutant Mice Regional Resource Center at the University of Davis, CA (*C57BL/6N-Lrfn2*$^{tm1(KOMP)Vlcg}$) and bred with C57BL/6J mice for more than six generations. Breeding cages were set up with heterozygous mice to obtain wild-type and *Lrfn2 KO* littermates for all experiments. *Vsx1-cerulean* (line 1) mice were a generous gift from Dr. Rachel Wong and bred as heterozygotes[37,68]. Dr. Vladimir Kefalov kindly provided *Gnat1 KO* mice on a C57BL/6J background[70]. Mice used in the study were between five days and four months of age. Postnatal day zero (P0) was defined as the day of birth. We found no differences in the results from mice of both sexes and, therefore, combined data acquired from them. Mice were housed on a 12-h light/12-h dark cycle. Behavioral experiments were conducted at 20–21 °C with 30–50% humidity levels. All procedures were approved by the Institutional Animal Care and Use Committee of Washington University School of Medicine (Protocol # 23-0116) and complied with

the National Institutes of Health Guide for the Care and Use of Laboratory Animals.

## Intraocular injections

For intraocular AAV injections, we anesthetized mouse pups (P6) on ice and injected 350 nL of *AAV9-Syn-iGluSnFR* (Addgene) into the vitreous chamber of both eyes with a Nanoject II injector (Drummond).

We anesthetized adult mice (P30-P60) with isoflurane for intraocular L-APB injections. We injected 1 μL of L-APB (100 μM) into the vitreous chamber of both eyes using a Nanoject II injector (Drummond). After the injections, mice were allowed to recover and dark-adapted for 30 min.

## Single-cell RNA-sequencing analysis

Data from Hahn et al.[48] was reanalyzed with Seurat (v5.0.3)[97]. Marker genes were used to identify rod and cone photoreceptors. We computed the average expression within each cell type and used ComplexHeatmap (v2.15.4)[98] and custom code to generate heatmaps. We visualized evolutionary relationships between species published by Hahn et al.[48] using ade4 (v1.7-22)[99].

## Retinal cDNA synthesis and target amplification

Retinas from wild-type and *Lrfn2 KO* mice were each processed for total RNA extraction using the RNeasy kit (Qiagen) according to the manufacturer's instructions. For each retina, 300 μg of total RNA was employed to generate cDNA in parallel reactions primed with oligo (dT) 12–18 or random hexamers using the SuperScript™ IV First-Strand Synthesis System (Invitrogen). The resulting cDNAs from each retina were then combined into a single pool (wild-type or *Lrfn2 KO*). PCR amplification of target genes was conducted with 2 μL of pooled cDNA per reaction, using KlenTaq polymerase under standard cycling conditions with a 56 °C annealing temperature. Because *Lrfn2* typically requires a higher annealing temperature (68 °C), its amplification under 56 °C conditions resulted in lower yield (Supplementary Fig. 5). Five microliters of each PCR product were resolved by gel electrophoresis, and a 1 kb GeneRuler™ (ThermoFisher Scientific) was used as the molecular weight marker.

## Immunohistochemistry

Retinal flat mounts and vertical vibratome sections were obtained as described previously[52]. After blocking for 1 h with 5% Normal Donkey Serum in PBS at room temperature (RT), vibratome slices (thickness: 60 μm) were incubated overnight at 4 °C with primary antibodies. Slices were then washed in PBS (3 × 20 min) and incubated with secondary antibodies for 2 h at RT. Flat-mount preparations were frozen and thawed three times after cryoprotection (1 h 10% sucrose in PBS at RT, 1 h 20% sucrose in PBS at RT, and overnight 30% sucrose in PBS at 4 °C), blocked with 5% Normal Donkey Serum in PBS for 2 h, and then incubated with primary antibodies for four days at 4 °C and washed in PBS (3 × 1 h) at RT. PNA-647 (Invitrogen) was added at 1:500 dilution together with the primary antibody. Flat mounts were incubated with the corresponding secondary antibodies for one day at 4 °C.

The following primary and secondary antibodies were used in this study: mouse anti-PKC (Sigma, 1:500, RRID: AB_477375), mouse anti-CtBP2 (BD Biosciences, 1:500, RRID:AB_399431), chicken anti-GFP (Invitrogen, 1:1000, RRID:AB_770014), rabbit anti-GluA1 (Millipore, 1:500, RRID:AB_2721164), rabbit anti-LRFN2 ([55] against N-terminus, 1:1000), rabbit anti-LRFN2 (against C-terminus, Sigma, 1:1000, RRID:AB_2138711), mouse anti-Bassoon (Abcam, 1:200, RRID: AB_1860018), mouse anti-Dystrophin (Santa Cruz, 1:250, RRID: AB_1122390), mouse anti-GluA2 (Invitrogen, 1:200, RRID: AB_2533058), mouse anti-GluK1 (Santa Cruz, 1:100, RRID: AB-2716684), mouse anti-Synaptotagmin 2 (Developmental Studies Hybridoma Bank, 1:200,

RRID: AB_2315626), sheep anti-mGluR6 (1:500, kind gift of Dr. Martemyanov[80]), Alexa 488-conjugated anti-chicken IgY anti-rabbit IgG, and Alexa 568-conjugated anti-mouse IgG, anti-goat IgG, and anti-sheep IgG (Invitrogen).

## Super-resolution imaging

We acquired super-resolution image stacks on a Zeiss LSM 880 with an Airyscan detector through a Plan-Apochromat 63x/1.4 Oil DIC M27 objective at a voxel size of 0.04 μm–0.14 μm (x/y – z). We processed image volumes in Amira (Visage Imaging) and ImageJ/Fiji (http://rsbweb.nih.gov/ij) and analyzed them with functions built into ImageJ/Fiji or custom scripts written in MATLAB (The Mathworks).

To measure cone pedicle volumes and active zone sizes, we individually masked cone arrestin (CAR)-stained cone pedicles in Amira. In addition to measuring the pedicle volume from the mask, we used it to isolate the PNA staining within a single pedicle. We then measured the active zone size by the area of the PNA signal in a maximum intensity projection above a threshold set by Otsu's method[100]. To analyze the colocalization of LRFN2 (*Signal2*) with different compartment-specific synapse markers (*Signal2*), we thresholded the respective signals using Otsu's method[100] and measured their *Overlap* as:

$$Overlap\,(\%) = \frac{2 \times (Signal1 \cap Signal2)}{Signal1 \cup Signal2} \times 100 \qquad (1)$$

## Patch-clamp electrophysiology

We recorded ON and OFF bipolar cells and transient OFFα ganglion cells by patch-clamp electrophysiology in retinal flat mounts. Retinas from dark-adapted (>1 h) mice were isolated under infrared illumination, mounted on poly-l-lysine-coated coverslips (Corning), and continually perfused (~7 ml/min) with warm (~32 °C) bicarbonate-buffered Ames medium with 95% $O_2$/5% $CO_2$. Recordings were performed using patch pipettes with tip resistance of 4–7 MΩ for ganglion cells and 10–12 MΩ for bipolar cells (borosilicate glass, WPI). Signals were amplified with a Multiclamp 700B amplifier (Molecular Devices), filtered at 3 kHz (8-pole Bessel low-pass), sampled at 10 kHz (Digidata 1440 A, Molecular Devices), and stored for analysis using pClamp 10 (Molecular Devices). In voltage-clamp recordings, series resistance (10–15 MΩ) was compensated electronically by ~75%. Excitatory and inhibitory postsynaptic currents (EPSCs and IPSCs) were isolated by holding cells at the reversal potential of inhibitory (−60 mV) and excitatory (0 mV) conductances, respectively. Voltages were corrected for a liquid junction potential of −10 mV. The intracellular solution for voltage-clamp recordings contained (in mM) 105 Cs-gluconate 20 Na-HEPES, 10 EGTA, 10 tetraethylammonium (TEA)-Cl, 2 Qx314, 5 adenosine 5′-triphosphate-Na, and 0.1 guanosine 5′-triphosphate-Na (285 mOsm, pH adjusted to 7.2 with CsOH). We targeted tOFFα ganglion cells and ON and OFF bipolar cells under infrared illumination. We confirmed cell identities by including Alexa 488 (0.1 mM) in the intracellular solution and acquiring two-photon image stacks at the end of each recording.

In patch-clamp recordings, visual stimuli were presented from an E4500 MKII PLUS II projector illuminated by a 385 nm light-emitting diode (EKB Technologies) and focused onto the photoreceptors of the retina via the substage condenser of an upright two-photon microscope. We attenuated the projector output with neutral density filters (Thorlabs). We measured the stimulus spectrum at the sample using a spectrometer (Black COMET, StellarNet). The mean intensity of visual stimuli in patch-clamp recordings was 1500 rhodopsin isomerization/rod/s (R*) and 314 S-opsin isomerizations/cone/s (S*). Stimuli were written

in MATLAB using Cogent graphics extensions (John Romaya, University College London, London, UK) and centered on the soma of the recorded cell. We measured light-evoked excitatory and inhibitory currents from bipolar cells to a square-wave-modulated spot of light (diameter: 60 μm, Michelson contrast: 100%, 2 s ON and 2 s OFF), determined the peak amplitudes (averaged across five stimulus repeats) during the ON and OFF phase of the stimulus and converted them into conductances based on the difference between the holding potential and the reversal potential of excitatory and inhibitory receptors.

## Two-photon imaging

We acquired images on a custom-built upright two-photon microscope (Scientifica) controlled by the Scanimage r3.8 MATLAB toolbox and recorded them with a DAQ NI PCI6110 data acquisition board (National Instruments). The genetically encoded glutamate sensor iGluSnFR was excited using an Insight DS+ laser (Spectra-Physics) tuned to 930 nm. The emitted fluorescence was collected through a 60 × 1.0 NA water immersion objective (Olympus) and filtered with consecutive 450 nm long-pass (Thorlabs, Newton, NJ) and 513–528 nm band-pass filters (Chroma, Bellows Falls, VT) to block the visual stimulus light (peak: 385 nm) from reaching the photomultiplier tube (PMT). The laser intensity at the sample was kept below 6 mW.

We isolated retinas from dark-adapted (>1 h) mice under infrared illumination in mouse artificial cerebrospinal fluid buffered with sodium bicarbonate (mACSF$_{NaHCO3}$) and flat mounted them on transparent membrane discs (Anodisc 13, Cytiva), which were secured in the microscope chamber. Throughout our imaging experiments, we superfused (3–7 mL/min) retinas with warm (31–33 °C) mACSF$_{NaHCO3}$ equilibrated with 95% O$_2$/5% CO$_2$.

We acquired images at 16.7 frames per second with a pixel density of 8.46 pixels/μm². To assess glutamate release from ON and OFF bipolar cells, we imaged iGluSnFR signals in scan fields across IPL depths in pseudorandom order. We measured IPL depths by their relative distance to the borders between the IPL and inner nuclear (0%) and ganglion cell layers (100%), respectively. For glutamate and calcium imaging, we allowed retinas to adapt to the laser light for 30 s before presenting visual stimuli. All images were acquired from the ventral retina, where S-opsin predominates in cones[101,102]. We applied L-APB (20 μM, Tocris) to hyperpolarize ON bipolar cells and isolate the function of the OFF bipolar pathway[103].

To register images in a time series, we simultaneously acquired fluorescence and transmitted light images. Based on the transmitted light images, we rejected time series with significant z-axis fluctuations. For time series passing this test, we registered images to the middle frame using built-in functions in MATLAB to apply matching rigid transformations to transmitted light and fluorescence images. The registration quality was confirmed by visual inspection before the transformed fluorescence images were used for further processing and analysis. Registered fluorescence images were median-filtered using a 3 × 3-pixel kernel. Visual stimuli and imaging time series were temporally aligned by detecting stimulus light with a temperature-compensated Si avalanche photodetector (Thorlabs), recorded with the imaging data via Scanimage r3.8.

We used a greedy correlation-based algorithm to segment the iGluSnFR time series into regions of interest (ROIs) representing bipolar cell axon boutons[33]. We generated ROI seeds by identifying the 60 pixels with the highest standard deviation over time. The signals of each seed pixel (starting with the highest-ranked) were correlated with the signals of all other pixels in the image. Connected pixels with a correlation coefficient ≥0.4 were assigned to the seed's ROI. ROIs were iteratively grown until the correlation of ROI-adjacent pixels with the seed was <0.4 or the ROI diameter exceeded 4 μm. ROIs with a diameter of <0.75 μm were removed from the final segmentation.

In two-photon imaging experiments, visual stimuli were presented from an E4500 MKII PLUS II projector illuminated by a 385 nm light-emitting diode (EKB Technologies) and focused onto the photoreceptors of the retina via the substage condenser of the microscope. We attenuated the projector output with neutral density filters (Thorlabs). We measured the stimulus spectrum at the sample using a spectrometer (Black COMET, StellarNet). The mean intensity of visual stimuli in two-photon imaging experiments was 8600 R* and 1800 S*. Stimuli were written in MATLAB using Cogent graphics extensions (John Romaya, University College London, London, UK). We presented a chirp stimulus to probe glutamate release from bipolar cells and calcium transients in ganglion cells. The chirp stimulus consists of a light step (1.5 s ON and 1.5 s OFF) followed by two sinusoidal intensity modulations: one with increasing frequency (0.5–40 Hz) at a fixed contrast (100%) and one with increasing contrast (10–100%) at a fixed temporal frequency (1 Hz). The chirp stimulus was presented in a spot (diameter: 100 μm) centered on the scan field and repeated thrice.

To analyze preferences for light increments vs. decrements, we calculated a polarity index based on responses to light steps at the beginning of the chirp stimulus (1.5 s ON, 1.5 s OFF) according to:

$$Polarity = \frac{ON - OFF}{ON + OFF} \tag{2}$$

where, ON and OFF represent the average responses during the respective phases of the stimulus.

To characterize the overall light responsiveness of an ROI, we measured the average correlation coefficient between its responses to three repeats of the chirp stimulus (i.e., reliability). Similarly, we measured the power of responses at the stimulus frequency during the chirp segment that increases contrast at a constant frequency (1 Hz) according to:

$$Power(rel.) = \frac{FFT_{1Hz}}{FFT_{Sum}} \tag{3}$$

where $FFT_{1Hz}$ indicates the response power at the stimulus frequency and $FFT_{Sum}$ the total power across the frequency spectrum calculated by fast Fourier transforms in MATLAB.

## Electroretinography

We dark-adapted mice overnight, anesthetized them with ketamine (0.1 mg/g body weight) and xylazine (0.01 mg/g body weight), and dilated their pupils dilated with 1% atropine sulfate (Falcon Pharmaceuticals). We recorded ERG responses in wild-type, *Lrfn2 KO*, *Gnat1 KO*, and *Gnat1 Lrfn2 DKO* mice using a UTAS Visual Electrodiagnostic Testing System (LKC Technologies). Recording electrodes embedded in contact lenses were placed over the cornea of both eyes. The mouse body temperature was maintained at 37 ± 0.5 °C throughout recordings with a heating pad controlled by a rectal temperature probe (FHC, Inc., Bowdoin, ME, USA). For flash ERGs, we averaged four to ten responses to 5 ms flashes at each light level, measured the a-wave as the difference between the response minimum in the first 50 ms after flash onset and the voltage value at flash onset, and the b-wave as the difference between a 15–25 Hz low to pass to filtered b-wave peak and the a-wave amplitude[52]. In flicker ERGs, we recorded responses to trains of brief flashes at 2.53 cdS/m² presented at varying rates (5, 7, 10, 12, 15, 18, 20, and 30 Hz) without background illumination[104]. Responses to flicker stimuli were mean-subtracted with a sliding window equal to one stimulus interval and averaged across 30 repeats before amplitudes were measured. All ERG analyses were performed using scripts written in MATLAB.

## Behavior

We evaluated looming-triggered innate defensive responses in a 45 × 27 × 31 cm box (width x depth x height) with three opaque walls and one transparent wall[10]. We recorded videos through the transparent wall with a USB camera (Logitech). We presented stimuli on an LCD monitor (display area: 32 × 24 cm, refresh rate: 60 Hz, mean stimulus intensity: 1350 R* and 740 S*), forming the box ceiling. Looming stimuli consisted of a 2° (diameter) dark disk on a gray background that expanded to 20° in 0.5 s and remained at this size for 0.25 s, before starting again at 2° for a sequence repeated 15 times without gaps. Mice were acclimatized to the behavioral arena for >5 min, and stimuli were started when mice entered the center of the arena.

Mouse positions, speeds, and freezing scores were analyzed using ANY-maze tracking software (Stoelting). Freezing was defined as a freezing score <30 to quantify the percentage of time frozen from stimulus onset to 10 s later[10].

## Statistics

Throughout this study, we evaluated the statistical significance of differences between experimental groups using Mann–Whitney $U$ tests. We compared flicker ERG frequency-responses functions by two-sided bootstrap tests.

## Reporting summary

Further information on research design is available in the Nature Portfolio Reporting Summary linked to this article.

## Data availability

All other data are available from the lead contacts, Florentina Soto (sotof@wustl.edu) and Daniel Kerschensteiner (kerschensteinerd@wustl.edu), upon request. Source data are provided with this paper.

## Code availability

The custom MATLAB scripts used for analysis are available at https://github.com/kerschensteinerd/soto_natcommun_2025 (https://doi.org/10.5281/zenodo.14975224).

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

## Acknowledgements

We thank members of the Kerschensteiner lab for helpful discussions throughout this study. This work was supported by the National Institutes of Health grants R01EY027411 (to F.S. and D.K.), R01EY026978 (to D.K.), R01EY034001 (to D.K.), and T32EY013360 (to C.-I.L.), the Grace Nelson Lacy Research Fund (to D.K.), the McDonnell Center for Systems Neuroscience (to D.K.), and an unrestricted grant from Research to Prevent Blindness (to the Department of Ophthalmology and Visual Sciences).

## Author contributions

F.S. and D.K. conceived of the project. C.-I.L. performed and analyzed two-photon glutamate imaging and behavior experiments. A.J. performed and analyzed patch-clamp experiments. S.-Y.C. and F.S. performed RT-PCR experiments. E.G.H. and P.A.R. analyzed single-cell RNA sequencing data. G.K.S. and R.S.P. contributed reagents. F.S. performed, and F.S. and D.K. analyzed all other experiments and wrote the manuscript with input from all authors.

## Competing interests

The authors declare no competing interests.
