## [Transparent Peer Review file · Nature Communications]

Molecular mechanism establishing the OFF pathway in vision

Corresponding Author: Dr Daniel Kerschensteiner

Version 0:

Reviewer comments:

Reviewer #1

(Remarks to the Author)

In the study by Soto et al., they used several new innovative approaches to show how a single cell adhesion molecule, Lrfrn2 is responsible for setting up the OFF visual pathway. The ON and OFF visual pathways are highly conserved among species, and establishing these distinct pathways are essential for normal vision. Although several molecules have been linked to the ON pathway, none have been reported to be selectively responsible for establishing the OFF pathway. The distinct visual pathways (i.e. OFF and ON) are set up at the first synapse between cone photoreceptors and their respective cone bipolar targets. In the OFF pathway, cones synapse to OFF cone bipolars, whereas in the ON pathway, cones synapse to ON cone bipolars. Soto and colleagues showed that Lrfrn2 is highly expressed in cone photoreceptors, and loss of Lrfrn2 disrupts cone-to-OFF cone bipolar connections but not cone-to-ON cone bipolars. They used super resolution imaging to show that Lrfrn2 is involved in clustering specific glutamate receptors (i.e. GluA1, GluK1) involved in OFF responses but not those that mediate ON responses (i.e. mGluR6). Moreover, they used patch-clamp recordings and two-photon glutamate imaging to show that indeed OFF responses were impaired in OFF cone bipolars but not those from ON bipolars. Furthermore, they were able to devise a new behavioral paradigm that allowed them to isolate the OFF responses in the looming stimuli task. Overall, this is a very interesting study that used several innovative tools to elucidate the molecular mechanism of how the OFF pathway is established in the mammalian retina.

Minor concerns:

1. Although the iGluSnFR experiments are quite interesting, it is hard to tell which OFF cone bipolar subtypes are affected in the Lrfrn2 KO. This would require using additional antibodies that label specific cone bipolar subtypes similar to anti-Syt2 that labels OFF cone bipolar subtype 2 as in the Supplemental data. However, these antibodies are quite finicky, and the data already supports that most OFF cone bipolars are affected based on the vsx1-cerulean mouse line in Fig 5.
2. Minor suggestion - the fluorescence intensity graphs in Fig 2b would be easier to read if they were shifted 90 degrees to match the fluorescent images and bar graphs.
3. GluK1 and GluA1 appear to still be expressed in Lrfrn2 KO and only mislocalized at the synapse. This could be confirmed by performing in situ hybridization to show that the mRNA transcript is still present at normal levels in both WT and Lrfrn2 KO animals.
4. Hard to see the dashed yellow lines that indicate the cone pedicles in Fig 4. Please make them slightly thicker.
5. It would be nice to show that the decrease in GluK1 and GluA1 expression is not due to loss of OFF cone bipolars. This could be addressed by counting the number of OFF cone bipolar subtype 2 in both WT and Lrfrn2 KO shown in Supp Fig 3.
6. Including the co-IP experiments mentioned in the Discussion section as Supplementary data would be good as this would show that Lrfrn2 does not directly bind to GluA1 and GluK1 receptors.

(Remarks on code availability)

Repository is empty.

Reviewer #2

(Remarks to the Author)

Previous studies have established that ELFN1 and ELFN2 are crucial for the formation of photoreceptor-to-ON bipolar cell (BC) synapses. However, the mechanisms underlying cone-to-OFF BC synapse formation remain poorly understood. In this study, Soto et al. identified LRFN2 as a potentially key molecule involved in this process, based on a multi-species scRNA-seq analysis of LRR-containing cell adhesion molecules in cones versus rods. The authors then investigated how LRFN2

influences various aspects of the OFF BC pathway, employing a combination of anatomical (immunolabeling and super-resolution imaging), physiological (patch-clamp and glutamate imaging), and behavioral assays in LRFN2 knockout (KO) mice. Their results suggest that LRFN2 plays a crucial role in cone-to-OFF BC synapse formation, pointing to the involvement of distinct LRR-containing cell adhesion molecules in the differential formation of cone-to-ON versus cone-to-OFF BC synapses. Overall, this is a systematic and thorough study that reveals an important mechanism in BC synapse formation. However, several points warrant further consideration:

1. While the authors convincingly demonstrate that LRFN2 is essential for cone-to-OFF BC synapse formation, the specific underlying mechanism remains unclear. The authors suggest that LRFN2 KO disrupts both pre- and postsynaptic membrane apposition as well as glutamate receptor clustering. However, one cannot rule out the possibility that the loss of receptor clustering is secondary to the disrupted membrane apposition, making it an indirect effect rather than a direct consequence of LRFN2 KO.
2. The authors should specify, or at least discuss, the types of OFF bipolar cells being analyzed in this study and consider the possibility of differential effects of LRFN2 on various OFF BC subtypes. It has been shown in multiple mammalian species that different OFF BC types contact distinct regions of cone pedicles, and these subtypes may express different compositions of KA and AMPA receptors. Notably, the iGluSnFR imaging data (Fig. 7j) suggest that OFF BCs stratifying more distally in the IPL are more affected by LRFN2 KO than those stratifying closer to the middle.
3. Since LRFN2 is involved in the synapse formation of all OFF BCs, LRFN2 knockout is expected to impact a broad range of RGC trigger features, rather than being specifically associated with looming detection. The authors hypothesize that looming detection may rely on rectified responses of OFF BCs. However, according to Fig. 7, the source of this rectification appears to stem from crossover inhibition from the ON pathway, rather than LRFN2-dependent input from cones.
4. Fig 7f, during L-APB treatment, glutamate release from the ON sublamina occurs during the OFF phase of light stimulation. Is this a delayed ON response, an OFF response, or an artifact due to misalignment?
5. Fig 5, scale bars are missing.

(Remarks on code availability)

Reviewer #3

(Remarks to the Author)

At the first synapse of the visual system, signals from photoreceptors diverge into two pathways, called ON and OFF, which respond to increments and decrements in light intensity, respectively. Several molecules have been implicated in the formation or maturation of synapses from photoreceptors to ON bipolar cells, but few if any have been found that play related roles at photoreceptor to OFF bipolar cell synapses. Here, in an unusually complete study, Soto et al. take a big step toward rectifying that imbalance. By reanalyzing published single cell RNA seq data, they chose LRFN2, an LRR repeat family transmembrane adhesion/recognition molecule that has been implicated in synaptic function and development elsewhere in the brain.

They localize the protein to the OFF compartment of cone terminals (pedicles; Figures 1 and 2 and supplement 2). They then analyze the knockout histologically showing decreased pedicle size (Figure 3), loss of OFF but not ON markers in the knockout (Figure 4 and Supplement 4), and displacement of OFF bipolar dendrites from pedicles (Figure 5) in the absence of LRFN2. Turning to physiology, they demonstrate decreased responsiveness of OFF but not ON bipolars to light in the knockout (Figures 6, 7, 8a-d and supplement 5). Finally, they show that a response to a visually threatening looming stimulus, which is known to be mediated (largely) through the OFF pathway, is decreased in the knockout (Figure 8f-j).

These results – which are presented in a logical progression to which my summary does not do justice – are well documented and convincing.

Data quality is high, but I have a few suggestions regarding the presentation:

1. Perhaps most important, Hasan and Gregg recently published a paper (2024, ref. 56) showing that LRFN2 is present in photoreceptors but claiming that it is necessary for function of the ON pathway. It seems that they never looked at the OFF pathway and misinterpreted their data about the ON pathway, so I have little doubt that Soto is correct and they are incorrect. However, Soto's excessive politeness could lead to confusion in the literature, so this issue needs to be discussed explicitly.
2. There is quite a literature on roles of LRFN2 in various brain circuits and synapses (just from the Hasan paper: Thevenon et al., 2016; Morimura et al., 2017; Li et al., 2018; Brouwer et al., 2019; McMillan et al., 2021). Some discussion of this work would be helpful, including comparison of previously reported phenotypes with the retinal ones. Likewise, it would be good to cite a few recent reviews that summarize what is known about the family (e.g., Liu H. *Curr Opin Struct Biol* and Lie E, Li Y, Kim R, Kim E. *Front Mol Neurosci*. 2018).
3. The one missing piece is the identity of the bipolar protein(s) with which LRFN2 interacts. Given the length and strength of this paper, I don't think it would be fair to ask for additional data, but it might be worth looking at available datasets to see if

there are plausible candidates that could be suggested. Papers by Shekhar (Cell) and Hahn (Nature) are already cited, but Sarin (Neuron, 2018) actually shows enrichment of LRFN2 in cones over rods and includes several other intriguing cis- and trans- possibilities.

4. The authors state that they have failed to find interactions of LRFN2 with glutamate receptors. They should, however, acknowledge that Zhou (Cancer Sci 2021) Maekawa (PlosOne 2021) and McMillan (eLife 2021) do show associations. Although the main emphasis has been on NMDARs, they are worth mentioning – and McMillan does focus on AMPARs.

5. The behavioral results are interesting but a bit complicated because LRFN2 affects a variety of brain circuits. This caution should be mentioned.

6. Is Supplement 1 from ref. 48? If so a reference/acknowledgement is needed in the legend.

(Remarks on code availability)

Reviewer #4

(Remarks to the Author)

(Remarks on code availability)

Version 1:

Reviewer comments:

Reviewer #1

(Remarks to the Author)

All concerns and suggestions have been fully addressed.

(Remarks on code availability)

Reviewer #2

(Remarks to the Author)

In this revision, the authors have adequately addressed my previous comments .

(Remarks on code availability)

Reviewer #3

(Remarks to the Author)

This is a terrific paper and my comments were minor. The additions to the revised manuscript address all of my concerns. I have also looked at the responses to Reviewers 1 and 3 and believe their requests - also fairly minor - have been answered satisfactorily.

(Remarks on code availability)

Reviewer #4

(Remarks to the Author)

(Remarks on code availability)

Reviewer #1:

In the study by Soto et al., they used several new innovative approaches to show how a single cell adhesion molecule, *Lrfn2* is responsible for setting up the OFF visual pathway. The ON and OFF visual pathways are highly conserved among species, and establishing these distinct pathways are essential for normal vision. Although several molecules have been linked to the ON pathway, none have been reported to be selectively responsible for establishing the OFF pathway. The distinct visual pathways (i.e. OFF and ON) are set up at the first synapse between cone photoreceptors and their respective cone bipolar targets. In the OFF pathway, cones synapse to OFF cone bipolars, whereas in the ON pathway, cones synapse to ON cone bipolars. Soto and colleagues showed that *Lrfn2* is highly expressed in cone photoreceptors, and loss of *Lrfn2* disrupts cone-to-OFF cone bipolar connections but not cone-to-ON cone bipolars. They used super resolution imaging to show that *Lrfn2* is involved in clustering specific glutamate receptors (i.e. GluA1, GluK1) involved in OFF responses but not those that mediate ON responses (i.e. mGluR6). Moreover, they used patch-clamp recordings and two-photon glutamate imaging to show that indeed OFF responses were impaired in OFF cone bipolars but not those from ON bipolars. Furthermore, they were able to devise a new behavioral paradigm that allowed them to isolate the OFF responses in the looming stimuli task. Overall, this is a very interesting study that used several innovative tools to elucidate the molecular mechanism of how the OFF pathway is established in the mammalian retina.

We appreciate the reviewer's astute summary and encouraging feedback on our work.

Minor concerns:

1. Although the iGluSnFR experiments are quite interesting, it is hard to tell which OFF cone bipolar subtypes are affected in the *Lrfn2* KO. This would require using additional antibodies that label specific cone bipolar subtypes similar to anti-Syt2 that labels OFF cone bipolar subtype 2 as in the Supplemental data. However, these antibodies are quite finicky, and the data already supports that most OFF cone bipolars are affected based on the *vsx1*-cerulean mouse line in Fig 5.

We thank the reviewer for raising this important point. Although our experiments do not allow us to distinguish the effects of *Lrfn2* KO on different OFF bipolar cell types, several lines of evidence (in addition to the dendritic dissociation from cones observed in the *Vsx1*-cerulean mice) indicate that LRFN2 controls synapse formation between cones and all OFF bipolar cell types (except GluMI¹ aka type 1B cells², which lack dendrites):

(1) L-APB attenuates light-evoked glutamate release from all OFF bipolar cell axon terminals in *Lrfn2* KO mice. Axons of different bipolar cell types stratify at distinct, overlapping depths in the IPL^{3,4}. We analyzed glutamate release from individual boutons across the IPL. In the presence of L-APB, light-evoked responses in *Lrfn2* KO mice were suppressed throughout the OFF sublamina. Moreover, in the presence of L-APB, we observed no boutons with well-entrained light responses. Given our extensive sampling, this strongly suggests that light responses of all OFF bipolar cell types are attenuated in these experiments. To better illustrate our findings, we have added three supplementary figures (Supplementary Figs. 9-11) showing the response traces at various IPL depths (Supplementary Fig. 9) and the distributions of response reliability (Supplementary Fig. 10) and power (Supplementary Fig. 11) of all ROIs (i.e., axonal boutons) at different IPL depths to our revised manuscript.

(2) All GluA1 and GluK1 clusters on OFF bipolar cell dendrites are lost in *Lrfn2 KO* mice. GluA1 and GluK1 cluster in the postsynaptic densities of most OFF bipolar cell types^{2,5,6}. In *Lrfn2 KO* mice, GluA1 and GluK1 staining is diffuse and displaced from the cone pedicle base. If some OFF bipolar cell types assembled their postsynaptic specializations independent of LRFN2, one might expect smaller receptor clusters to remain in their normal positions in *Lrfn2 KO* mice. Thus, our immunohistochemical observations (in addition to the dendritic dissociation from cones in *Vsx1-cerulean* mice) support the notion that LRFN2 controls synapse formation between cones and all OFF bipolar cell types.

(3) Excitatory input to OFF bipolar cells is reduced uniformly in *Lrfn2 KO* mice. By patch-clamp electrophysiology, light-evoked EPSCs were attenuated in all OFF bipolar cells we recorded in *Lrfn2 KO* retinas (n = 7 cells). Unlike the two-photon glutamate imaging experiments, sampling in our patch-clamp experiments is not exhaustive. However, it included OFF bipolar cells with diverse axonal stratification patterns, likely reflecting diverse OFF bipolar cell types (**Reviewer Fig. 1**).

Taken together, these observations suggest that LRFN2 is essential for normal synapse formation between cones and all OFF bipolar cell types with dendrites. A more detailed assessment of how *Lrfn2 KO* affects individual OFF bipolar cell types and of potential cell-type-specific variations in LRFN2's contributions remains an important direction for future work. We have added a paragraph clarifying our interpretation to the Discussion section of our revised manuscript.

2. Minor suggestion - the fluorescence intensity graphs in Fig 2b would be easier to read if they were shifted 90 degrees to match the fluorescent images and bar graphs.

We appreciate this suggestion. We tried rotating the fluorescence intensity graphs by 90° but found this orientation was not optimal. Because the cone pedicles and the corresponding representative images are wider than they are tall, rotating the graphs compressed the critical axis that shows the laminar positioning of LRFN2 relative to other synaptic markers. To clarify the relationship between the image

orientation and the intensity profile, we now include a distance axis in the first representative image (panel a) of our revised **Fig. 2**. We hope this addition will help our readers more intuitively follow the conversion from the vertical axis in the image to the horizontal axis in the plot.

3. GluK1 and GluA1 appear to still be expressed in *Lrfn2* KO and only mislocalized at the synapse. This could be confirmed by performing in situ hybridization to show that the mRNA transcript is still present at normal levels in both WT and *Lrfn1* KO animals.

Following the reviewer's recommendations, we analyzed the mRNA expression of *Grik1* (GluK1) and *Gria1* (GluA1). RT-PCR revealed no apparent changes in glutamate receptor expression levels between wild-type and *Lrfn2* KO mice, indicating that LRFN2 controls the molecular assembly of the synapse, not the expression of its components. We include the results of our RT-PCR in a new supplementary figure (**Supplementary Fig. 5**) in our revised manuscript.

4. Hard to see the dashed yellow lines that indicate the cone pedicles in Fig 4. Please make them slightly thicker.

We have increased the line thickness as the reviewer suggested.

5. It would be nice to show that the decrease in GluK1 and GluA1 expression is not due to loss of OFF cone bipolars. This could be addressed by counting the number of OFF cone bipolar subtype 2 in both WT and *Lrfn2* KO shown in Supp Fig 3.

Following the reviewer's suggestion, we performed additional analysis to test if OFF bipolar cells are lost in *Lrfn2* KO mice. We measured the density of type 3B OFF bipolar cells (stained for PKARII β) and all nuclei in the INL (labeled with DAPI). Both revealed no significant differences between wild-type and *Lrfn2* KO mice. Together with the mRNA expression data mentioned in our response to the reviewer's comment #3 (**Supplementary Fig. 5**), these results support the interpretation that the reduced GluK1 and GluA1 staining results from a failure of these receptors to cluster in the OFF bipolar cell postsynapse. We have included the analysis of INL and type 3B OFF bipolar cell numbers in a new supplementary figure (**Supplementary Fig. 6**) in our revised manuscript.

6. Including the co-IP experiments mentioned in the Discussion section as Supplementary data would be good as this would show that *Lrfn2* does not directly bind to GluA1 and GluK1 receptors.

We thank the reviewer for this comment. We do not want to draw strong conclusions from our failure to detect co-immunoprecipitation of GluA1 and GluK1 with LRFN2. Therefore, we choose not to include the respective data in our manuscript. Instead, we include it here for the reviewer's benefit (**Reviewer Fig. 2**). We hope the reviewer is okay with this choice.

Remarks on code availability:

Repository is empty.

We apologize for this oversight. We have now added the code to the GitHub repository for this manuscript (https://github.com/kerschensteinerd/soto_natcommun_2025).

Reviewer #2:

Previous studies have established that ELFN1 and ELFN2 are crucial for the formation of photoreceptor-to-ON bipolar cell (BC) synapses. However, the mechanisms underlying cone-to-OFF BC synapse formation remain poorly understood. In this study, Soto et al. identified LRFN2 as a potentially key molecule involved in this process, based on a multi-species scRNA-seq analysis of LRR-containing cell adhesion molecules in cones versus rods. The authors then investigated how LRFN2 influences various aspects of the OFF BC pathway, employing a combination of anatomical (immunolabeling and super-resolution imaging), physiological (patch-clamp and glutamate imaging), and behavioral assays in LRFN2 knockout (KO) mice. Their results suggest that LRFN2 plays a crucial role in cone-to-OFF BC synapse formation, pointing to the involvement of distinct LRR-containing cell adhesion molecules in the differential formation of cone-to-ON versus cone-to-OFF BC synapses. Overall, this is a systematic and thorough study that reveals an important mechanism in BC synapse formation. However, several points warrant further consideration:

We are grateful for the reviewer's perceptive summary and positive feedback on our study.

1. While the authors convincingly demonstrate that LRFN2 is essential for cone-to-OFF BC synapse formation, the specific underlying mechanism remains unclear. The authors suggest that LRFN2 KO disrupts both pre- and postsynaptic membrane apposition as well as glutamate receptor clustering. However, one cannot rule out the possibility that the loss of receptor clustering is secondary to the disrupted membrane apposition, making it an indirect effect rather than a direct consequence of LRFN2 KO.

We thank the reviewer for highlighting this important point. We find that LRFN2 is required (1) for the adhesion of OFF bipolar cell dendrites to the cone pedicle based and (2) for the clustering of glutamate receptors in the postsynaptic membrane. Our experiments do not allow us to distinguish whether these functions of LRFN2 are mechanistically identical (e.g., LRFN2 interacts trans-synaptically with the iGluR complex to mediate cell adhesion and stabilize receptor clusters), independent (e.g., LRFN2 engages in distinct trans-synaptic interactions to mediate cell adhesion and receptor clustering), or whether one phenotype in *Lrfn2 KO* mice is secondary to the other (both orders would be possible).

To address the reviewer's comment, we outline the mechanisms that could underlie the observed phenotypes in *Lrfn2 KO* mice (as outlined in the previous paragraph) and highlight the need for further mechanistic studies in the Discussion section of our revised manuscript.

2. The authors should specify, or at least discuss, the types of OFF bipolar cells being analyzed in this study and consider the possibility of differential effects of LRFN2 on various OFF BC subtypes. It has been shown in multiple mammalian species that different OFF BC types contact distinct regions of cone pedicles, and these subtypes may express different compositions of KA and AMPA receptors. Notably, the iGluSnFR imaging data (Fig. 7j) suggest that OFF BCs stratifying more distally in the IPL are more affected by LRFN2 KO than those stratifying closer to the middle.

We thank the reviewer for raising the issue of OFF bipolar cell diversity. Although our experiments do not allow us to distinguish the effects of *Lrfn2 KO* on different OFF bipolar cell types, several lines of

evidence indicate that LRFN2 controls synapse formation between cones and all OFF bipolar cell types (except GluMI¹ aka type 1B cells², which lack dendrites):

(1) In the presence of L-APB, light-evoked glutamate release from OFF bipolar cell axon terminals in *Lrfn2 KO* mice is uniformly suppressed across the OFF sublamina. Because different OFF bipolar cell types stratify at distinct yet overlapping depths in the IPL^{3,4}, we analyzed glutamate release from individual axonal boutons at multiple depths and found no significant differences in the attenuated responses. Moreover, no boutons exhibited robust, phase-locked light responses under L-APB treatment. This broad suppression strongly suggests that *Lrfn2 KO* impairs synaptic function in all OFF bipolar cell types. To illustrate these findings, we have added three supplementary figures (**Supplementary Figs. 9-11**) showing representative traces at various IPL depths (**Supplementary Fig. 9**) and the distributions of response reliability (**Supplementary Fig. 10**) and power (**Supplementary Fig. 11**) for all ROIs.

(2) All GluA1 and GluK1 clusters on OFF bipolar cell dendrites are lost in *Lrfn2 KO* mice. GluA1 and GluK1 cluster in the postsynaptic densities of most OFF bipolar cell types in mice^{2,5,6}. In *Lrfn2 KO* retinas, GluA1 and GluK1 staining is diffuse and displaced from the cone pedicle base. If some OFF bipolar cell types assembled their postsynaptic specializations independent of LRFN2, one might expect smaller receptor clusters to remain in their normal positions in *Lrfn2 KO* mice. Thus, our immunohistochemical observations (in addition to the dendritic dissociation from cones in *Vsx1-cerulean* mice) support the notion that LRFN2 controls synapse formation between cones and all OFF bipolar cell types.

(3) Excitatory input to OFF bipolar cells is reduced uniformly in *Lrfn2 KO* mice. By patch-clamp electrophysiology, light-evoked EPSCs were attenuated in all OFF bipolar cells we recorded in *Lrfn2 KO* retinas (n = 7 cells). Unlike the two-photon glutamate imaging experiments, sampling in our patch-clamp experiments is not exhaustive. However, it included OFF bipolar cells with diverse axonal stratification patterns, likely reflecting diverse OFF bipolar cell types (see **Reviewer Fig. 1**, included our response to Reviewer #1).

Taken together, these observations suggest that LRFN2 is essential for normal synapse formation between cones and all OFF bipolar cell types with dendrites. A more detailed assessment of how *Lrfn2 KO* affects individual OFF bipolar cell types and of potential cell-type-specific variations in LRFN2's contributions remains an important direction for future work. We have added a paragraph clarifying our interpretation to the Discussion section of our revised manuscript.

3. Since LRFN2 is involved in the synapse formation of all OFF BCs, LRFN2 knockout is expected to impact a broad range of RGC trigger features, rather than being specifically associated with looming detection. The authors hypothesize that looming detection may rely on rectified responses of OFF BCs. However, according to Fig. 7, the source of this rectification appears to stem from crossover inhibition from the ON pathway, rather than LRFN2-dependent input from cones.

We thank the reviewer for identifying this mistake. Indeed, the persistence of normal defensive responses of mice intraocularly injected with L-APB indicates that looming detection is independent of the OFF pathway rectification. We have rewritten the respective section of the discussion in our revised manuscript

as follows: “The dedicated retinal OFF pathway has also been hypothesized to aid the detection of approaching predators. Consistent with this hypothesis, we find that *LRFN2* shapes the responses of *tOFFa* ganglion cells and the innate defensive behaviors relying on these responses (Fig. 8)^{7,8}.”

4. Fig 7f, during L-APB treatment, glutamate release from the ON sublamina occurs during the OFF phase of light stimulation. Is this a delayed ON response, an OFF response, or an artifact due to misalignment?

We thank the reviewer for noting the glutamate signals in the ON sublamina during the OFF phase of light stimulation in the presence of L-APB. The short, fixed latency of these signals indicates that they are not delayed ON responses, unlike those previously observed in ganglion cell spikes under ON-pathway impairment^{9,10}. Instead, our analysis of signals across IPL depths shows that these OFF-phase responses are strongest in the center of the IPL and weaken toward the ganglion cell layer boundary. Moreover, they are attenuated in *Lrfn2 KO* mice, suggesting that this phenomenon represents glutamate spillover from OFF bipolar cells. Because iGluSnFR has a higher affinity for glutamate than iGluRs, detecting these signals does not imply that spillover is functionally significant.

We provide extended data on glutamate signals at different IPL depths (under both L-APB and control conditions) in three new supplementary figures (**Supplementary Figs. 9-11**) and further explain the OFF responses in the ON sublamina in the Results section of our revised manuscript.

5. Fig 5, scale bars are missing.

We thank the reviewer for spotting this omission. We have added scale bars to Fig. 5 in our revised manuscript.

Reviewer #3:

At the first synapse of the visual system, signals from photoreceptors diverge into two pathways, called ON and OFF, which respond to increments and decrements in light intensity, respectively. Several molecules have been implicated in the formation or maturation of synapses from photoreceptors to ON bipolar cells, but few if any have been found that play related roles at photoreceptor to OFF bipolar cell synapses. Here, in an unusually complete study, Soto et al. take a big step toward rectifying that imbalance. By reanalyzing published single cell RNA seq data, they chose LRFN2, an LRR repeat family transmembrane adhesion/recognition molecule that has been implicated in synaptic function and development elsewhere in the brain.

They localize the protein to the OFF compartment of cone terminals (pedicles; Figures 1 and 2 and supplement 2). They then analyze the knockout histologically showing decreased pedicle size (Figure 3), loss of OFF but not ON markers in the knockout (Figure 4 and Supplement 4), and displacement of OFF bipolar dendrites from pedicles (Figure 5) in the absence of LRFN2. Turning to physiology, they demonstrate decreased responsiveness of OFF but not ON bipolars to light in the knockout (Figures 6, 7, 8a-d and supplement 5). Finally, they show that a response to a visually threatening looming stimulus, which is known to be mediated (largely) through the OFF pathway, is decreased in the knockout (Figure 8f-j).

These results – which are presented in a logical progression to which my summary does not do justice – are well documented and convincing.

We thank the reviewer for their insightful summary and positive remarks.

Data quality is high, but I have a few suggestions regarding the presentation:

1. Perhaps most important, Hasan and Gregg recently published a paper (2024, ref. 56) showing that LRFN2 is present in photoreceptors but claiming that it is necessary for function of the ON pathway. It seems that they never looked at the OFF pathway and misinterpreted their data about the ON pathway, so I have little doubt that Soto is correct and they are incorrect. However, Soto's excessive politeness could lead to confusion in the literature, so this issue needs to be discussed explicitly.

We thank the reviewer for encouraging us to clarify the commonalities and distinctions between both studies. Accordingly, we have added the following paragraph to the Discussion section of our revised manuscript:

“Our findings align with Hasan and Gregg¹¹, who also located LRFN2 at the cone pedicle base. However, while Hasan and Gregg¹¹ focused on the ON pathway—finding subtle changes in the ERG b-wave—we detected no differences in the ERG (Supplementary Fig. 8) or synaptic inputs from cones to ON bipolar cells in Lrfn2 KO mice (Fig. 5 and Supplementary Fig. 7). Instead, our anatomical, physiological, and behavioral data (Figs. 2 and 4-8) reveal that LRFN2 is specifically required for synapses between cones and OFF bipolar cells, a pathway that Hasan and Gregg¹¹ did not investigate.”

2. There is quite a literature on roles of LRFN2 in various brain circuits and synapses (just from the Hasan paper: Thevenon et al., 2016; Morimura et al., 2017; Li et al., 2018; Brouwer et al., 2019; McMillan et al., 2021). Some discussion of this work would be helpful, including comparison of previously reported phenotypes with the retinal ones. Likewise, it would be good to cite a few recent reviews that summarize what is known about the family (e.g., Liu H. *Curr Opin Struct Biol* and Lie E, Li Y, Kim R, Kim E. *Front Mol Neurosci*. 2018).

We thank the reviewer for encouraging us to extend our discussion to compare the phenotypes we observe in the retinas of *Lrfn2 KO* mice to those observed elsewhere in their nervous system. We have added the following paragraph to the Discussion section of our revised manuscript, including some of the citations highlighted by the reviewer:

*“In the hippocampus (and other brain areas), LRFN2 is found in the postsynaptic densities of excitatory synapses^{12, 13, 14}. There is debate about the deficits in hippocampal synapse development of *Lrfn2 KO* mice^{15, 16}, but overall, phenotypes of LRFN2 disruption appear milder in the hippocampus than we observed in the retina, affecting similar domains (e.g., receptor clustering, synaptic adhesion, and synapse morphology)^{12, 14, 15, 16}. Milder phenotypes may indicate compensation by other mechanisms. The opposite localization of LRFN2 (retina: presynaptic, hippocampus: postsynaptic) matches observations for other cell-adhesion molecules and components of the intracellular synaptic scaffold (including PSD95)^{17, 18, 19, 20}. The purpose of this partially inverse molecular architecture of the photoreceptor synapse remains to be fully understood.”*

3. The one missing piece is the identity of the bipolar protein(s) with which LRFN2 interacts. Given the length and strength of this paper, I don't think it would be fair to ask for additional data, but it might be worth looking at available datasets to see if there are plausible candidates that could be suggested. Papers by Shekhar (*Cell*) and Hahn (*Nature*) are already cited, but Sarin (*Neuron*, 2018) actually shows enrichment of LRFN2 in cones over rods and includes several other intriguing cis- and trans- possibilities.

See our combined response to points #3 and 4 below

4. The authors state that they have failed to find interactions of LRFN2 with glutamate receptors. They should, however, acknowledge that Zhou (*Cancer Sci* 2021) Maekawa (*PlosOne* 2021) and McMillan (*eLife* 2021) do show associations. Although the main emphasis has been on NMDARs, they are worth mentioning – and McMillan does focus on AMPARs.

Following the reviewer's recommendation (points #3 and #4), we have added the following paragraph to the Discussion section of our revised manuscript to outline potential molecular mechanisms by which LRFN2 could control synapse formation with OFF bipolar cells. We have included the citations highlighted by the reviewer:

*“Whether LRFN2 mediates synaptic adhesion with OFF bipolar cells and postsynaptic receptor clustering independently or if one deficit in *Lrfn2 KO* mice results from the other (e.g., glutamate receptors diffuse because synaptic adhesions are lost) remains to be explored. Similarly, the molecular mechanisms by which LRFN2 exerts trans-synaptic control over OFF bipolar cells are unknown. No*

trans-synaptic interaction partners of LRFN2 have yet been identified^{21, 22}. In cultured cells, the extracellular region of LRFN2 can interact with NMDA and AMPA receptors in cis (i.e., in the same membrane)^{13, 14, 16, 23, 24}, and in the ON pathway, ELFN1 and ELFN2 interact trans-synaptically with mGluR6^{18, 25, 26}. While it is possible that LRFN2 at the cone pedicle base interacts trans-synaptically with AMPA and kainate receptors in OFF bipolar cells, we did not detect such interactions by co-immunoprecipitation (data not shown). Alternatively, LRFN2 could control synaptic adhesion and postsynaptic receptor clustering through yet-unidentified trans-synaptic partners^{2, 27, 28} or, indirectly, through presynaptic actions²⁹.”

5. The behavioral results are interesting but a bit complicated because LRFN2 affects a variety of brain circuits. This caution should be mentioned.

We thank the reviewer for raising this critical point. Although we cannot rule out contributions of LRFN2 downstream of the retina, we think they are unlikely to dominate the behavioral result because (1) *Lrfn2* does not seem to be expressed in the brain areas that mediate looming responses (see **Reviewer Fig. 3**) and (2) the behavioral effect depends on retinal L-APB injections. To acknowledge the uncertainty and explain our interpretation, we have added the following paragraph to the Results section of our revised manuscript:

“We cannot rule out that LRFN2 influences looming responses through visual circuits downstream of the retina. However, the limited expression of Lrfn2 in the respective brain areas (including the superior colliculus, parabigeminal nucleus, and dorsal periaqueductal gray) and the effect of intraocular L-APB injections argue against this^{12, 30, 31, 32}.”

6. Is Supplement 1 from ref. 48? If so a reference/acknowledgement is needed in the legend.

We thank the reviewer for spotting this omission. We have added the reference to the respective figure legend.

Reviewer #4:

We thank you for participating in the review and for your input.

References

1. Della Santina L, *et al.* Glutamatergic Monopolar Interneurons Provide a Novel Pathway of Excitation in the Mouse Retina. *Curr Biol* **26**, 2070-2077 (2016).
2. Shekhar K, *et al.* Comprehensive Classification of Retinal Bipolar Neurons by Single-Cell Transcriptomics. *Cell* **166**, 1308-1323.e1330 (2016).
3. Franke K, Berens P, Schubert T, Bethge M, Euler T, Baden T. Inhibition decorrelates visual feature representations in the inner retina. *Nature* **542**, 439-444 (2017).
4. Helmstaedter M, Briggman KL, Turaga SC, Jain V, Seung HS, Denk W. Connectomic reconstruction of the inner plexiform layer in the mouse retina. *Nature* **500**, 168-174 (2013).
5. Wässle H, Puller C, Müller F, Haverkamp S. Cone contacts, mosaics, and territories of bipolar cells in the mouse retina. *J Neurosci* **29**, 106-117 (2009).
6. Borghuis BG, Looger LL, Tomita S, Demb JB. Kainate receptors mediate signaling in both transient and sustained OFF bipolar cell pathways in mouse retina. *J Neurosci* **34**, 6128-6139 (2014).
7. Kim T, Shen N, Hsiang JC, Johnson KP, Kerschensteiner D. Dendritic and parallel processing of visual threats in the retina control defensive responses. *Science Advances* **6**, eabc9920 (2020).
8. Wang F, Li E, De L, Wu Q, Zhang Y. OFF-transient alpha RGCs mediate looming triggered innate defensive response. *Curr Biol*, (2021).
9. Demas J, *et al.* Failure to maintain eye-specific segregation in nob, a mutant with abnormally patterned retinal activity. *Neuron* **50**, 247-259 (2006).
10. Renteria RC, Tian N, Cang J, Nakanishi S, Stryker MP, Copenhagen DR. Intrinsic ON responses of the retinal OFF pathway are suppressed by the ON pathway. *J Neurosci* **26**, 11857-11869 (2006).
11. Hasan N, Gregg RG. Cone Synaptic function is modulated by the leucine rich repeat (LRR) adhesion molecule LRFN2. *eNeuro* **11**, 2023.2005.2024.542135 (2024).
12. Ko J, *et al.* SALM synaptic cell adhesion-like molecules regulate the differentiation of excitatory synapses. *Neuron* **50**, 233-245 (2006).
13. McMillan KJ, *et al.* Sorting nexin-27 regulates AMPA receptor trafficking through the synaptic adhesion protein LRFN2. *Elife* **10**, (2021).

14. Wang C-Y, Chang K, Petralia RS, Wang Y-X, Seabold GK, Wenthold RJ. A novel family of adhesion-like molecules that interacts with the NMDA receptor. *J Neurosci* **26**, 2174-2183 (2006).
15. Li Y, *et al.* Lrfn2-Mutant Mice Display Suppressed Synaptic Plasticity and Inhibitory Synapse Development and Abnormal Social Communication and Startle Response. *J Neurosci* **38**, 5872-5887 (2018).
16. Morimura N, *et al.* Autism-like behaviours and enhanced memory formation and synaptic plasticity in Lrfn2/SALM1-deficient mice. *Nat Commun* **8**, 15800 (2017).
17. Haverkamp S, Grünert U, Wässle H. The Cone Pedicle, a Complex Synapse in the Retina. *Neuron* **27**, 85-95 (2000).
18. Cao Y, *et al.* Mechanism for Selective Synaptic Wiring of Rod Photoreceptors into the Retinal Circuitry and Its Role in Vision. *Neuron* **87**, 1248-1260 (2015).
19. Martemyanov KA, Sampath AP. The Transduction Cascade in Retinal ON-Bipolar Cells: Signal Processing and Disease. *Annu Rev Vis Sci* **3**, 25-51 (2017).
20. Burger CA, Jiang D, Mackin RD, Samuel MA. Development and maintenance of vision's first synapse. *Dev Biol* **476**, 218-239 (2021).
21. Nam J, Mah W, Kim E. The SALM/Lrfn family of leucine-rich repeat-containing cell adhesion molecules. *Semin Cell Dev Biol* **22**, 492-498 (2011).
22. Lie E, Li Y, Kim R, Kim E. SALM/Lrfn Family Synaptic Adhesion Molecules. *Front Mol Neurosci* **11**, 105 (2018).
23. Zhou Y, *et al.* LRFN2 binding to NMDAR inhibits the progress of ESCC via regulating the Wnt/beta-Catenin and NF-kappaB signaling pathway. *Cancer Sci* **113**, 3566-3578 (2022).
24. Maekawa R, Muto H, Hatayama M, Aruga J. Dysregulation of erythropoiesis and altered erythroblastic NMDA receptor-mediated calcium influx in Lrfn2-deficient mice. *PLoS One* **16**, e0245624 (2021).
25. Cao Y, *et al.* Interplay between cell-adhesion molecules governs synaptic wiring of cone photoreceptors. *Proc Natl Acad Sci U S A* **117**, 23914-23924 (2020).
26. Dunn HA, Patil DN, Cao Y, Orlandi C, Martemyanov KA. Synaptic adhesion protein ELFN1 is a selective allosteric modulator of group III metabotropic glutamate receptors in trans. *Proc Natl Acad Sci U S A* **115**, 5022-5027 (2018).

27. Hahn J, *et al.* Evolution of neuronal cell classes and types in the vertebrate retina. *Nature* **624**, 415-424 (2023).
28. Sarin S, *et al.* Role for Wnt Signaling in Retinal Neuropil Development: Analysis via RNA-Seq and In Vivo Somatic CRISPR Mutagenesis. *Neuron* **98**, 109-126 e108 (2018).
29. Brouwer M, *et al.* SALM1 controls synapse development by promoting F-actin/PIP2-dependent Neurexin clustering. *EMBO J* **38**, e101289 (2019).
30. Shang C, *et al.* BRAIN CIRCUITS. A parvalbumin-positive excitatory visual pathway to trigger fear responses in mice. *Science* **348**, 1472-1477 (2015).
31. Evans DA, Stempel AV, Vale R, Ruehle S, Lefler Y, Branco T. A synaptic threshold mechanism for computing escape decisions. *Nature* **558**, 590-594 (2018).
32. Lein ES, *et al.* Genome-wide atlas of gene expression in the adult mouse brain. *Nature* **445**, 168-176 (2007).

Reviewer #1 (Remarks to the Author):

All concerns and suggestions have been fully addressed. (Remarks on code availability)

We thank the reviewer for the positive assessment of our revisions.

Reviewer #2 (Remarks to the Author):

In this revision, the authors have adequately addressed my previous comments. (Remarks on code availability)

We thank the reviewer for the positive assessment of our revisions.

Reviewer #3 (Remarks to the Author)

This is a terrific paper and my comments were minor. The additions to the revised manuscript address all of my concerns. I have also looked at the responses to Reviewers 1 and 3 and believe their requests - also fairly minor - have been answered satisfactorily. (Remarks on code availability)

We thank the reviewer for the positive assessment of our revisions.

Reviewer #4 (Remarks to the Author)

I co-reviewed this manuscript with one of the reviewers who provided the listed reports. This is part of the Nature Communications initiative to facilitate training in peer review and to provide appropriate recognition for Early Career Researchers who co-review manuscripts.(Remarks on code availability)

We thank the reviewer for their contributions to the review of our manuscript.